# Thalamic reticular control of local sleep in mouse sensory cortex

**Laura MJ Fernandez, Gil Vantomme, Alejandro Osorio-Forero, Romain Cardis, Elidie Béard, Anita Lüthi\***

Department of Fundamental Neurosciences, University of Lausanne, Lausanne, Switzerland

**Abstract** Sleep affects brain activity globally, but many cortical sleep waves are spatially confined. Local rhythms serve cortical area-specific sleep needs and functions; however, mechanisms controlling locality are unclear. We identify the thalamic reticular nucleus (TRN) as a source for local, sensory-cortex-specific non-rapid-eye-movement sleep (NREMS) in mouse. Neurons in optogenetically identified sensory TRN sectors showed stronger repetitive burst discharge compared to non-sensory TRN cells due to higher activity of the low-threshold $Ca^{2+}$ channel $Ca_V3.3$. Major NREMS rhythms in sensory but not non-sensory cortical areas were regulated in a $Ca_V3.3$-dependent manner. In particular, NREMS in somatosensory cortex was enriched in fast spindles, but switched to delta wave-dominated sleep when $Ca_V3.3$ channels were genetically eliminated or somatosensory TRN cells chemogenetically hyperpolarized. Our data indicate a previously unrecognized heterogeneity in a powerful forebrain oscillator that contributes to sensory-cortex-specific and dually regulated NREMS, enabling local sleep regulation according to use- and experience-dependence.

DOI: https://doi.org/10.7554/eLife.39111.001

## Introduction

Sleep is a global vigilance state with well-known behavioral, electroencephalographic and neuromodulatory attributes. However, cerebral correlates of non-rapid-eye-movement sleep (NREMS) and REMS, notably several major EEG sleep rhythms, occur variably at different times in different brain regions (*Massimini et al., 2004*; *Nir et al., 2011*; *Siclari and Tononi, 2017*). This suggests that, on top of a global regulation, forebrain pacemakers with regionally specific oscillatory properties shape sleep across the cortex (*Krueger et al., 2013*). Such local aspects probably underlie the sleeping brain's decreased capacity to integrate external information and enable plasticity in specific neural circuits (*Siclari and Tononi, 2017*; *Crunelli et al., 2018*). For example, sleep-dependent memory consolidation is linked to spatially confined regulation of NREMS rhythms in the brain areas involved in recent learning (*Rasch and Born, 2013*). Furthermore, sleep disorders may arise from a pathologically altered spatial heterogeneity that negatively impacts sleep as a global state (*Krueger et al., 2013*; *Siclari and Tononi, 2017*).

Prototype cellular pacemakers for NREMS rhythms, notably for the slow-oscillation (SO) (<1 Hz), delta waves (1–4 Hz) and sleep spindles (10–15 Hz) have been known for decades (for review, see (*Steriade et al., 1993*; *Astori et al., 2013*; *Sanchez-Vives et al., 2017*)). However, the spatiotemporal variability of cortical rhythms challenges the idea that these are homogeneous sources across the cortical surface (*Piantoni et al., 2016*; *Siclari and Tononi, 2017*). For example, 'fast' and 'slow' human sleep spindles distribute variably and correlate differentially with memory consolidation, which has prompted a search for at least two, if not several, separately active spindle generators (*Schabus et al., 2007*; *Frauscher et al., 2015*). Interestingly, anatomical and functional boundaries of cortical areas go in parallel with variations of sleep rhythms (*Fernandez et al., 2017*;

**\*For correspondence:**
anita.luthi@unil.ch

**Competing interests:** The authors declare that no competing interests exist.

**eLife digest** Falling asleep affects our behavior immediately and profoundly. During sleep, large electrical waves appear across the brain in areas responsible for consciousness, sensation and movement. In the cortex – the outer layer of the brain – sleep waves arise from networks that connect to the thalamus, a deeper structure within the brain. However, not all areas of the brain sleep equally. We know this intuitively because sensory stimuli, such as an alarm clock or a baby's cry, can still wake us up. By contrast, we typically do not move much or take major decisions while we sleep. Therefore, the brain areas involved in sensation should not be expected to sleep in the same way as areas involved in movement or reasoning.

Neighboring brain areas generally show very different sleep waves. The brain regions that we use during the day can also affect how sleep varies from one area to the next. It is not well understood what determines these 'local' sleep properties.

By studying the brains of mice, Fernandez et al. now show that the networks between the cortex and thalamus are much more varied than previously thought, in particular regarding a thalamic nucleus that is relevant for sleep wave generation. These previously unrecognized differences deep within the brain are part of the origin of local sleep in the outer layer of the brain. Sleep wave activity differed depending on whether the networks were involved in sensory or non-sensory roles. The networks allow sensory areas to switch efficiently between different forms of local sleep. This might underlie how the brain's sensory activity during the day can influence local sleep at night.

There is growing evidence that major sleep disorders are due to disturbances to local sleep. Techniques to modify or restore specific sleep waves locally in the brain could help to develop new sleep therapies. For example, having a detailed map of electrical waves within the sleep-disordered brain could help researchers to apply transcranial stimulation techniques in ways that might help to treat these debilitating disorders.

DOI: https://doi.org/10.7554/eLife.39111.002

*Piantoni et al., 2017*). Moreover, sleep rhythms can show local singularities within a single cortical area according to developmental stage, or as a result of experience and learning during the day (*Huber et al., 2004*; *Kurth et al., 2010*; *Johnson et al., 2012*; *Laventure et al., 2016*). There is also evidence for the localized appearance of sleep-related patterns in individual cortical columns (*Pigarev et al., 1997*; *Rector et al., 2005*). Therefore, modality-specific thalamocortical loops could account for diversity in local NREMS. Moreover, there should be powerful local tuning mechanisms to locally switch between different NREMS rhythms. The current view is dominated by cortical focal influences as essential to shape local rhythms (*Contreras et al., 1996*; *Piantoni et al., 2017*; *Siclari and Tononi, 2017*). In contrast, thalamic oscillators are seen as broad and relatively homogeneous sources of oscillatory power that can spread focally or globally to cortex (*Bonjean et al., 2012*), but that have little bearing on their ultimate cortical correlates.

This view is becoming revised as novel observations on heterogeneous thalamic pacemaker mechanisms are reported. Of interest is the thalamic reticular nucleus (TRN), typically referred to as an inhibitory shell of highly oscillatory, burst-prone cells surrounding the dorsal thalamus (*Pinault, 2004*; *Fogerson and Huguenard, 2016*). Burst discharge generates large inhibitory synaptic potentials (*Herd et al., 2013*; *Rovó et al., 2014*) that entrain thalamocortical neurons into rhythmicity. Bursting is based on the $Ca_V3.3$ channel that is crucial for sleep spindle generation (*Astori et al., 2011*; *Pellegrini et al., 2016*), but many studies indicate that not all TRN cells burst equally (*Contreras et al., 1992*; *Brunton and Charpak, 1997*; *Lee et al., 2007*; *Kimura et al., 2012*; *Clemente-Perez et al., 2017*; *Higashikubo and Moore, 2018*). Then, the TRN is parcellated into at least five modality-specific sectors, sensory, motor or limbic ones, according to their innervation by a particular dorsal thalamic nucleus and the reciprocally connected cortical area (*Crabtree, 1999*; *Pinault, 2004*). Moreover, optogenetic activation of TRN promotes cortical spindles or delta waves (*Halassa et al., 2011*; *Lewis et al., 2015*), suggesting that the exact patterns of TRN cell activity may determine the contribution of these two rhythms at the cortical surface. Most recently, various discharge propensity in TRN cells was linked to the differential expression of parvalbumin (PV) or somatostatin (*Clemente-Perez et al., 2017*). The number of PV-expressing cells is

smaller in TRN of schizophrenic patients and mouse models (*Steullet et al., 2018*) in which reduced sleep spindle density is a common observation (*Manoach et al., 2016*). Together, there is accruing evidence for marked molecular and functional TRN cell heterogeneity. However, whether TRN heterogeneity is relevant for brain correlates of NREMS has so far not been tested.

This study shows that the heterogeneous cellular properties of optogenetically identified TRN sectors are a major source of local NREMS rhythms in sensory cortices. We identify the ionic mechanisms underlying this heterogeneity and study its impact on major NREMS correlates in sensory and non-sensory cortices using genetic and chemogenetic approaches. We thereby unravel novel organizing principles of NREMS topography in mouse and show that heterogeneity in TRN sectors accounts for a previously unrecognized enrichment of fast and large sleep spindles in somatosensory cortices that are coupled to the SO. We also find that TRN heterogeneity enables rapid switching between different forms of NREMS rhythms, suggesting that this could underlie the regulation of local NREMS by use and experience.

## Results

### TRN cell burst discharge propensity in acute slices varies across sensory and non-sensory sectors

To identify TRN cells belonging to a sensory sector, we used stereotaxic injections of AAV-ChR2_EYFP into somatosensory (S1, barrel field) or auditory cortex (AC) of 3- to 4-week-old mice. Non-sensory TRN sectors related to associative areas, such as the medial prefrontal cortex (PFC), were instead targeted through injections into the mediodorsal (MD) thalamic nucleus that is the input structure forming reciprocal loops with several areas of the PFC (*Mátyás et al., 2014*; *Delevich et al., 2015*; *Collins et al., 2018*) and that forms reciprocal circuits with the TRN (*Mitchell, 2015*). Enhanced yellow fluorescent protein (EYFP) fluorescence was present at injection sites and in restricted portions of TRN 3–4 weeks after injection, as verified on coronal sections stained

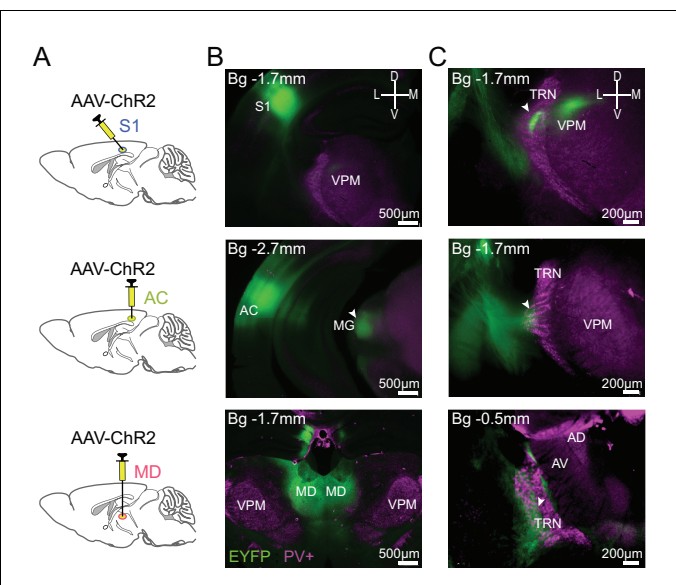

**Figure 1.** Identification of TRN sectors through anterograde tracing. (**A**) Scheme illustrating injection of AAV-ChR2_EYFP into S1 (top), AC (middle) and MD (bottom). (**B**) Epifluorescent micrographs of coronal sections at 2.5x magnification, anteroposterior position indicated with respect to bregma (Bg). ChR2-EYFP (green) expression in infected S1 neurons and their projection to TRN and VPM (top), in AC neurons and their projection to medial geniculate nucleus (MG) (middle) and in MD neurons (bottom), with immunostaining for PV-positive (PV+, magenta) TRN neurons. The VPM appears in light magenta due to its innervation by TRN. (**C**) Expanded view of the target TRN sector at 5x magnification. White arrowheads indicate sites of projection into TRN.
DOI: https://doi.org/10.7554/eLife.39111.003

immunohistochemically for PV to delineate the dorso-ventral extent of the TRN (*Figure 1*). The observed sectorial portions coincided with the ones established previously (*Pinault and Deschênes, 1998*; *Pinault, 2004*). Thus, injections into S1 revealed fibers navigating through the postero-dorsal portion of the TRN that terminated in elongated fluorescent spots in the thalamic ventral posterior medial nucleus corresponding to thalamic barreloids (*Bourassa et al., 1995*). AC injections resulted in fluorescent labeling of postero-central regions of the TRN that are anterior to the medial geniculate nucleus. Finally, MD injections labeled antero-ventral portions of the TRN, overlapping with TRN areas innervating motor and intralaminar nuclei (*Pinault and Deschênes, 1998*; *Pinault, 2004*).

In acute coronal slices prepared from injected animals, TRN cells recorded in the green fluorescent areas through whole-cell patch clamp recordings reliably (> 85 % of the cells across sectors) responded to optogenetic stimulation (470 nm light, pulse duration $\leq$1 ms, 0.16 mW/mm$^2$) with rapid excitatory postsynaptic currents ranging between -42 to -938 pA (*Figure 2A*). Paired stimuli (interstimulus interval: 100 ms) yielded paired-pulse facilitation for cortical afferents (S1-innervated TRN cells: 203 $\pm$ 12%, n = 12, Wilcoxon signed rank-test, p=4.9x10$^{-4}$ for 2$^{nd}$ vs 1$^{st}$ stimulus; for AC-innervated TRN cells: 175 $\pm$ 21%, n = 6, p=0.03), as described previously (*Astori and Lüthi, 2013*), whereas paired responses to MD stimulation showed comparable size (105 $\pm$ 4 %, n = 13, p=0.27). Cells showed values for resting membrane potential and capacitance consistent with previous data (*Figure 2B*; *Astori et al., 2011*). Rebound action potential discharge was elicited in response to square somatic current injections (negative current injections of -50 to -300 pA for 500 ms to hyperpolarize the somatic membrane potential < -100 mV, yielding cell input resistance values of 344 $\pm$ 18 M$\Omega$ for all three sectors). Rebound oscillatory bursting hallmarks the capacity of TRN cells to engage in rhythm generation (*Astori et al., 2011*; *Wimmer et al., 2012*; *Clemente-Perez et al., 2017*). S1- and AC-innervated TRN cells showed the rhythmic, repetitive burst discharge described previously (*Figure 2C*; *Cueni et al., 2008*; *Astori et al., 2011*), evident as several groups of high-frequency action potentials each riding on a triangular-shaped membrane depolarization and followed by a pronounced afterhyperpolarization. Repetitive burst discharge strongly depended on the membrane voltage, showing an inverted U-shaped voltage dependence that peaked at -65 to -60 mV for S1-innervated cells (*Figure 2D1,E*). Only 1/12 S1-innervated TRN cells was a non-repetitive bursting cell (*Figure 2F*). Similar burst propensity was found for TRN cells innervated from AC (*Figure 2E*), but 4/13 cells were non-repetitive bursters (*Figure 2F*). In TRN cells responding to MD inputs, repetitive bursting was weak (*Figure 2C,D1*) and 9/14 cells discharged maximally one burst (*Figure 2F*). These results show that repetitive burst propensity is stronger in sensory compared to non-sensory TRN sectors. Within sensory sectors, the somatosensory sector displayed the highest density of strongly bursting cells, whereas the auditory sector had a smaller proportion of cells with rhythmic bursting.

To test whether the heterogeneity of burst discharge across TRN sectors depended on Ca$_V$3.3 channels, the optogenetic strategy described above was applied to animals with a genetic deletion of the *Cacna1i (Ca$_V$3.3, $\alpha$1I)* gene (*Astori et al., 2011*). This channel is primarily responsible for burst discharge in TRN cells, while co-expressed Ca$_V$3.2 channels play a minor role (*Pellegrini et al., 2016*). Accordingly, TRN cells of these animals are unable to burst repetitively, whereas tonic action potential discharge is preserved. Cells patched in acute slices from *Cacna1i$^{-/-}$* (Ca$_V$3.3 KO) animals showed passive properties comparable to those in C57BL/6J (WT) cells of the corresponding sectors (*Figure 2B*), although S1-innervated cells had a smaller capacitance indicative of reduced cell size, perhaps resulting from morphological alterations in these constitutive knock-outs. Light-evoked synaptic responses showed a similar range of amplitudes ($-10$ to $-1076$ pA) and a short-term plasticity that was comparable to the one found for WT cells (S1-innervated TRN cells: 188 $\pm$ 32%, n = 7, p=0.016; for AC-innervated TRN cells: 203 $\pm$ 13%, n = 8, p=0.008; for MD-innervated TRN cells: 87 $\pm$ 8, n = 10, p=0.31; two-way ANOVA with factors 'genotype' and 'sector', p=0.28 for interaction). However, the vigorous bursting in somatosensory and auditory TRN cells was suppressed, thus abolishing the dependence of repetitive burst discharge propensity on TRN sector type (Kruskal-Wallis with factor 'sector', p=3$\times$10$^{-4}$ for WT, p=0.14 for Ca$_V$3.3 KO) (*Figure 2D2, E*). Together, these data indicate that the Ca$_V$3.3 channel endows superior bursting capacity to sensory over non-sensory TRN cells.

It has been shown that TRN bursting capacities are sensitive to cortical lesions (*Paz et al., 2010*). To exclude the possibility that the viral injections compromised TRN discharge, we also recorded from TRN cells in slices prepared from uninjected animals and identified putative sensory and non-

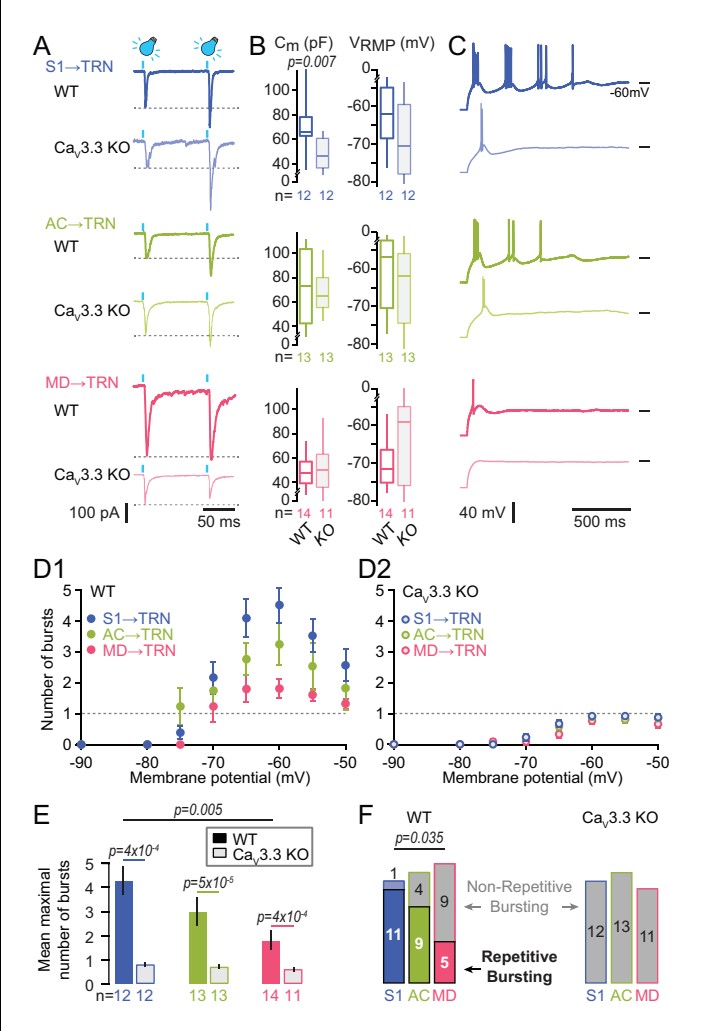

**Figure 2.** Oscillatory burst firing varies across TRN sectors and depends on Ca$_V$3.3 Ca$^{2+}$ channels. (A) Representative traces of paired EPSCs in whole-cell voltage-clamped TRN neurons at −60 mV of WT and Ca$_V$3.3 KO mice upon light activation of S1 (top), AC (middle) and MD (bottom) afferents. Afferent-specific forms of short-term plasticity are preserved across genotypes (see Results for averaged data). (B) Box-and-whisker-plots of the mean capacitance (C$_m$) and the resting membrane potential (V$_{RMP}$) of the recorded TRN neurons (WT: n = 12 for S1, n = 13 for AC, n = 14 for MD; Ca$_V$3.3 KO: n = 12 for S1, n = 13 for AC, n = 11 for MD). It can be noted that the C$_m$ of S1-innervated TRN neurons in Ca$_V$3.3 KO showed a reduction compared to WT, suggesting smaller cell size (Mann-Whitney test, p=0.007). (C) Representative current-clamp recordings of oscillatory bursting responses of TRN neurons across sectors, induced through hyperpolarizing current injections (−50 to −300 pA for 500 ms). Horizontal lines denote −60 mV. Note the strong repetitive burst firing in sensory sectors that is impaired in the Ca$_V$3.3 KO cells, whereas MD-innervated cells mostly discharge a single burst. (D) Graph of the number of repetitive bursts as a function of the membrane potential prior to the hyperpolarizing pulse (*Cueni et al., 2008*). This yields a U-shaped curve reaching a peak at −65 and −60 mV in all sectors of WT mice (D1) that was abolished in Ca$_V$3.3 KO mice (D2). Dashed horizontal lines at ordinate value one indicate the border between repetitive and not-repetitive bursting conditions. (E) Mean number of repetitive bursts of TRN neurons (between −60 and −65 mV) across sectors and genotype. Mann-Whitney tests were used for comparison between genotypes, and p-values are given above the bars. (F) Histogram of the proportion of repetitive (colored rectangles) and non-repetitive bursting (grey rectangles with color surroundings) TRN neurons in the different sectors. Chi-square test followed by pairwise proportion test with Holm's p-value adjustment was used for statistical evaluation, with significant value given above the bars.

DOI: https://doi.org/10.7554/eLife.39111.004

The following source data and figure supplement are available for figure 2:

**Source data 1.** Numerical data values and statistics underlying **Figure 2**.

*Figure 2 continued on next page*

*Figure 2 continued*

DOI: https://doi.org/10.7554/eLife.39111.006

**Figure supplement 1.** Oscillatory burst firing varies across the anteroposterior extent of TRN and depends on Ca$_V$3.3 Ca$^{2+}$ channels.

DOI: https://doi.org/10.7554/eLife.39111.005

sensory sectors in horizontal slices along the anteroposterior axis. These experiments confirmed the different burst propensity in sensory vs non-sensory sectors (*Figure 2—figure supplement 1*).

## Mouse NREMS shows spectral features that are specific to functional cortical areas

We next monitored local NREMS in cortical areas connected to the TRN sectors studied *in vitro*. Under stereotaxic guidance, animals were implanted for *in vivo* multi-site recordings of local field potentials (LFPs) in the same cortical areas that were previously targeted for the anterograde viral tracing of TRN sectors, along with electroencephalography/electromyography (EEG/EMG) (*Figure 3A,B*). For S1 and AC, electrodes were positioned in deep layers (layers 5 and 6), whereas infra-/prelimbic cortical areas (collectively referred to as PFC) were implanted in middle layers (layers 3 and 5, see Materials and methods for exact stereotaxic coordinates), according to the majority of thalamocortical input received in the respective areas. Additionally, the secondary somatosensory cortex (S2) was implanted (layers 2/3 and 4) to monitor NREMS in an associative sensory cortical area with strong reciprocal connections to S1 (*Zingg et al., 2014*). We chose high-impedance electrodes (~10–12 MΩ) for LFP recordings to maximize detection of local signals. Simultaneous EEG/EMG recordings on the contralateral hemisphere were used for vigilance state scoring (*Figure 3A, B*).

Animals were recorded in head-restrained mode, which yields a sleep profile comparable to that in freely moving conditions, as previously shown (*Fernandez et al., 2017*; *Lecci et al., 2017*). Each mouse was recorded for 2–3 hr/day and spontaneously switched between periods of wakefulness, NREMS and REMS with power spectra typical for each vigilance state (*Figure 3—figure supplement 1A*).

NREMS was accompanied by distinct LFP waveforms across cortical areas of WT animals (*Figure 3C*). In S1 and S2, prominent activity in the SO (0.5–1.5 Hz, frequency band chosen based on visual inspection of the power spectrum) and the sleep spindle (sigma, 10–15 Hz) frequency ranges was visible (*Figure 3—figure supplement 1B*). Activity in the delta (1.5–4 Hz) frequency range was evident as large positive deflections (*Figure 3—figure supplement 1B*). Similar, yet weaker rhythmic activity was observed in AC and PFC. The PFC showed signals dominated by slow events, as reported (*Fernandez et al., 2017*), which included a component around 4 Hz resembling a respiratory-related rhythm in frontal brain areas (*Figure 3C,D*; *Zhong et al., 2017*). Power spectral analyses over total NREMS times of 2200–6100 s per animal (average 4487 ± 603 s, concatenated from NREMS bouts across recording days) showed that NREMS in all recorded areas had broadly elevated power in the low-frequency range covering both the SO and the delta range (0.5–4 Hz), whereas a 'shoulder' in the sigma band was present only in somatosensory areas (n = 9 for S1 and n = 8 for S2) but not in AC (n = 6) and PFC (n = 6) (*Figure 3D*).

NREMS in animals lacking Ca$_V$3.3 channels showed several marked changes that were apparent in both the raw traces and in characteristic alterations of the power spectra (NREMS recording times of 1800–7000 s per animal, average 3459 ± 421 s). First, there was a striking lack of visually recognizable sleep spindle activity in S1 and S2, and a sigma power shoulder was not present in the power spectrum (*Figure 3C,D*, bottom panel showing enlarged portions of the power spectrum and *Figure 3—figure supplement 1B*). Second, activity in the delta range was augmented, whereas the SO was less prominent in LFP traces from S1, S2 and AC. These visual observations manifested as a rightward shift of the low-frequency activity in the power spectrum, with a clear power peak present in the delta band that dominated over power values < 1.5 Hz. The power spectra for WT and Ca$_V$3.3 KO animals intersected in the slow frequencies around 1.6–1.8 Hz for S1, S2 and AC, suggesting a consistent spectral border between the SO and the delta bands. Separate quantification of total power in the SO, the delta and the sigma frequency band confirmed these observations (*Figure 3E*). After Bonferroni correction, the sigma power reduction in S1 appeared as a trend. However, this is

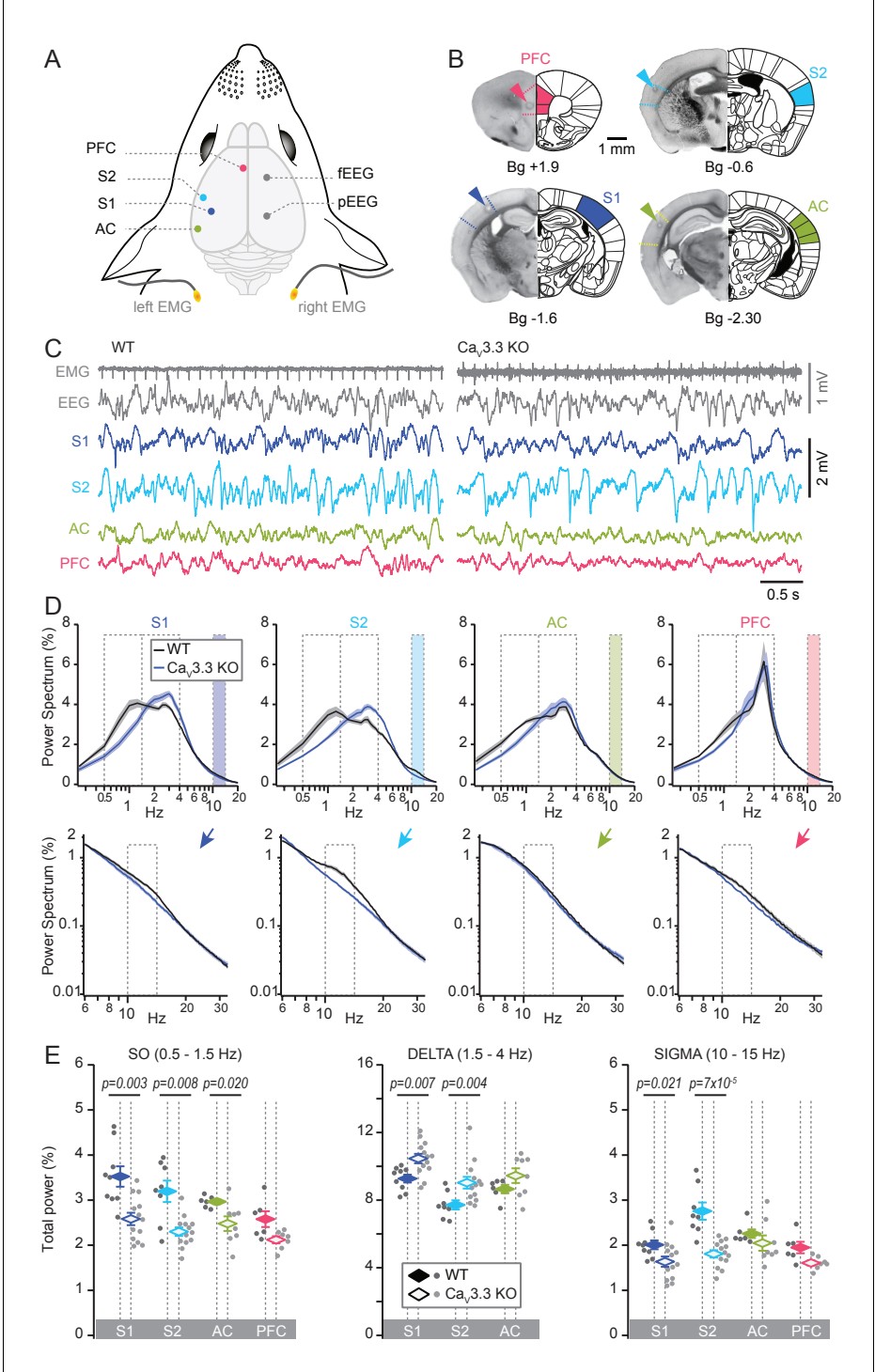

**Figure 3.** Cortical-area specific NREMS features depend on Ca$_V$3.3 Ca$^{2+}$ channels. (**A**) Schematic illustrating implantation sites for LFP, EEG (fEEG and pEEG; frontal and parietal EEG) and EMG electrodes. S1, S2; primary and secondary somatosensory areas; AC, auditory cortex; PFC, medial prefrontal cortex. (**B**) Histological sections of representative cases confirming the location of the recording sites. Arrowheads mark site of lesion caused by electrocoagulation. Anteroposterior stereotaxic coordinates are given relative to Bregma (Bg). (**C**) Representative raw traces of NREMS for WT (left) and Ca$_V$3.3 KO (right) animals, showing (from top to bottom) the EMG, EEG and LFP signals for S1, S2, AC and PFC (infra-/prelimbic area). The heart beat is visible on the EMG trace. (**D**) Power spectra corresponding to the LFP recordings, plotted in a linear-log plot to emphasize the three frequency bands of interest: the SO (0.5–1.5 Hz), the delta (1.5–4 Hz) and the sigma band (10–15 Hz). The sigma band is colored and

*Figure 3 continued on next page*

*Figure 3 continued*

shown in expanded log-log plots at the bottom. Normalized mean ±S.E.M. values of power spectral density are shown for S1, S2, AC and PFC for both genotypes (WT: n = 9 for S1, n = 8 for S2, n = 6 for AC, n = 6 for PFC; $Ca_V3.3$ KO: n = 13 for S1, n = 13 for S2, n = 8 for AC, n = 7 for PFC). (E) Mean total power for the three frequency bands across S1, S2, AC and PFC, with values for individual animals shown in points (dark gray for WT, light gray for $Ca_V3.3$ KO animals), and mean values ± S.E.M. in color diamonds. Statistical significance was tested for each area separately, comparing WT and $Ca_V3.3$ KO. Mann-Whitney non-parametric test for WT *vs* $Ca_V3.3$ KO, for S1, p=0.003 for the SO, p=0.007 for delta, p=0.021 for sigma; for S2, p=0.008 for the SO, p=0.005 for delta, p=$6.9\times10^{-5}$ for sigma; for AC, p=0.02 for the SO, p>0.05 for delta and sigma; for PFC, all p-values>0.05. Bonferroni-corrected α-threshold for the three frequency bands was 0.017.

DOI: https://doi.org/10.7554/eLife.39111.007

The following source data and figure supplement are available for figure 3:

**Source data 1.** Numerical data values and statistics underlying *Figure 3*.
DOI: https://doi.org/10.7554/eLife.39111.009
**Figure supplement 1.** Head-fixed animals present spectra typical for each vigilance state accompanied by distinct LFP waveforms across cortical areas in WT and $Ca_V3.3$ KO animals.
DOI: https://doi.org/10.7554/eLife.39111.008

an underestimation because the difference between the two curves extends up to ~18 Hz. In PFC, in contrast, no alterations in the power spectrum were observed, with in particular no significant change in the sigma power and in the SO peak. The delta band was not analyzed in this area due to the superposition of the respiratory rhythm on top of the delta waves.

Together, the lack of $Ca_V3.3$ channels altered the spectral mix of NREMS in sensory circuits, whereby power in the delta frequency band became overrepresented compared to the SO. This shift was greatest in S1 and S2, where in addition sigma power activity was suppressed. In contrast, the PFC did not show these alterations in two of the three major frequency bands, suggesting a minor dependence on $Ca_V3.3$ channels.

## Chemogenetic inhibition of TRN cells reproduces the switch from spindle- to delta-enriched sleep

The results from $Ca_V3.3$ KO animals suggest that strong $Ca_V3.3$-dependent burst discharge is required for spindle-enriched NREMS. Therefore, acute reduction of TRN excitability to suppress bursting should also deplete spindles and lead to a NREMS enriched in delta waves. To test this, we hyperpolarized TRN cells with a chemogenetic approach, whereby we expressed the inhibitory DREADD receptor hM4Di in VGAT-Ires-Cre animals (*Vong et al., 2011*) through bilateral injection of AAV-hM4D(Gi)_mCherry or AAV-hM4D(Gi)_IRES_mCitrine in the region of the somatosensory TRN sector (*Figure 4—figure supplement 1A*). In acute slices prepared 3 weeks after injection, bath application of the DREADD-ligand Clozapine N-oxide (CNO, 10 μM) induced a marked membrane hyperpolarization (ΔV = −13.9 ± 1.5 mV, n = 10, paired *t*-test, p=$6\times10^{-6}$) of fluorescent cells held in whole-cell current-clamp at resting membrane potentials ranging from −50 to −70 mV (*Figure 4A, B*). Cellular input resistance was reduced and rebound burst discharge suppressed in the continuous presence of CNO. Bursting could be recovered upon direct current (d.c.) injection to restore the original membrane potential (d.c. injection tested in n = 3 cells, *Figure 4A*). Non-fluorescent cells in the vicinity of the injected area did not respond to CNO (*Figure 4B*). This result is consistent with CNO-induced activation of $K^+$ conductances and shows that TRN neurons become silenced without impairing rebound bursting and action potential firing. Treatment with CNO thus overall reduces TRN excitability, and in particular specifically reproduces the decreased burst propensity of TRN cells found in the $Ca_V3.3$-KO animals that is relevant for the altered NREMS spectral properties.

Similarly injected animals were also implanted *in vivo* for S1 LFP and EEG/EMG freely moving recordings and treated with CNO (i.p. 1 mg/kg) or NaCl at 2 hr into the light phase (ZT2). The latencies to fall asleep were comparable after drug or NaCl injections (31.4 ± 3.7 min for CNO, 24.8 ± 2.1 min for NaCl; n = 5, paired *t*-test, p=0.16). NREMS analysis was done for the time period of 20–65 min after drug injection, which is the period where drug effects peak (*Figure 4—figure supplement 1B*). Total time spent in NREMS was not different between CNO and NaCl injections in the analysis period (24.1 ± 1.7 min for CNO, 27.7 ± 1.9 min for NaCl injections, Wilcoxon signed rank-test,

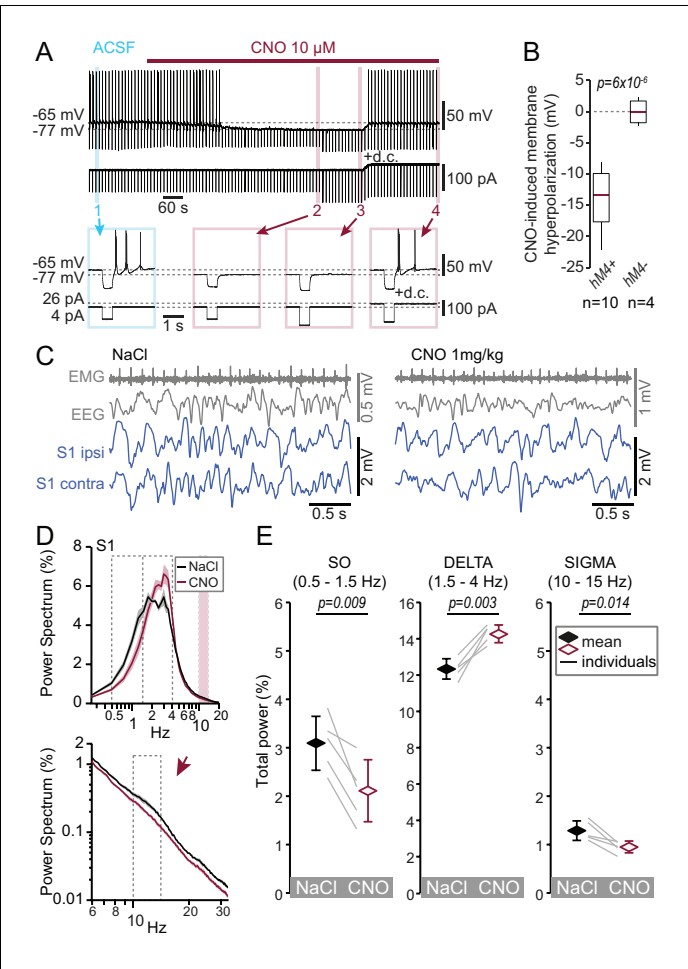

**Figure 4.** Chemogenetic hyperpolarization of TRN cells mimics the NREMS phenotype of the Ca$_V$3.3 KO mice. (**A**) Representative membrane voltage recording of a whole-cell patch-clamped TRN cell *in vitro* exposed to CNO (10 μM, bath application indicated by horizontal bar) recorded in a slice from a mouse injected with AAV8-hM4D(Gi) _IRES_mCitrine. The cell was injected every 10 s with brief negative current pulses to elicit rebound discharge. The application of CNO hyperpolarized the membrane potential, suppressed rebound bursting and decreased membrane input resistance, as evident by the smaller voltage deflection in response to the negative current step (−80 pA). Current step size was increased to −120 pA to compensate for the decreased membrane resistance. Subsequent injection of direct current (d.c.) to counteract membrane hyperpolarization then reinstated burst discharge. Numbers indicate portions of the trace shown expanded below. Horizontal dotted lines indicate mean membrane potential before and during CNO application. (**B**) Box-and-whisker plot of membrane hyperpolarization *in vitro* (ΔV, calculated as the difference before and during CNO) for fluorescent (hM4+, n = 10, ΔV = −13.9 ± 1.5 mV, paired *t*-test, p=6×10$^{-6}$) and non-fluorescent cells (hM4-, n = 4, ΔV = 0.0 ± 0.9 mV, paired *t*-test, p=0.97). The CNO-effects between the two-cell groups differed significantly (unpaired *t*-test, p=1.1×10$^{-4}$). (**C**) Representative traces *in vivo* during NREMS 30 min after the injection of NaCl (left) or CNO (right) in the same animal, showing (from top to bottom) the EMG, EEG and S1-LFP (ipsilateral and contralateral to EEG) signals. (**D**) Mean ±S.E.M. power spectra of the S1-LFPs for NaCl and CNO injections *in vivo* during the NREMS periods 20 to 65 min after injection. Expanded portion is shown below in log-log scale to emphasize the sigma band (10–15 Hz). (**E**) Mean total power for the three frequency bands of interest. Diamonds and error bars show the Mean ±S.E.M. across subjects. Gray lines represent individual animals. Repeated-measures ANOVA for factors 'frequency' and 'treatment', p=7.7×10$^{-5}$ was followed by paired *t*-tests for individual frequency bands, with values given above the bars. Bonferroni-corrected, α threshold was 0.017.

DOI: https://doi.org/10.7554/eLife.39111.010

The following source data and figure supplement are available for figure 4:

**Source data 1.** Numerical data values and statistics underlying *Figure 4*.

DOI: https://doi.org/10.7554/eLife.39111.012

*Figure 4 continued on next page*

*Figure 4 continued*

**Figure supplement 1.** Chemogenetic inhibition of TRN cells acutely increases delta activity in a DREADD-dependent expression.

DOI: https://doi.org/10.7554/eLife.39111.011

p=0.22) and mean NREMS bout durations were comparable (93.3 ± 12.3 s for CNO, 91.2 ± 32.5 s for NaCl, Wilcoxon signed rank-test, p=0.31). Following CNO injections, S1 LFP signals during NREMS showed reduced spindle activity and instead became enriched in activity in the delta frequency range (*Figure 4C,D*, repeated-measures ANOVA with factors 'frequency' and 'treatment', p=7.7×10$^{-5}$). Compared to NaCl injections, total power in the delta frequency range was increased, whereas the SO and spindle activity were suppressed (*Figure 4D,E*). Control animals injected with AAV8 carrying a DREADD-unrelated optogenetic construct (see Materials and methods) did not respond to CNO (*Figure 4—figure supplement 1C–F*). The CNO-induced acute membrane hyperpolarization thus reproduced the major power spectral changes observed in NREMS of the Ca$_V$3.3 KO mouse: the suppression of the SO and sleep spindle power, and the enhancement of delta power. As CNO-induced hyperpolarization suppressed burst discharge (see also *Figure 2D1*), the joint results from the genetic and the chemogenetic manipulations identify decreased TRN bursting as the primary factor relevant for the enrichment of delta power at the expense of sigma and SO power in NREMS.

## Ca$_V$3.3 channels amplify and accelerate spindles in the somatosensory areas

Given the importance of Ca$_V$3.3-dependent TRN burst discharge for NREMS in sensory cortices, the question remains open of how this discharge pattern controls the properties of local, discrete spindle events. The exact sources of regional spindle properties have recently received considerable attention (*Schabus et al., 2007*; *Frauscher et al., 2015*; *Piantoni et al., 2017*). Therefore, we developed an algorithm to isolate discrete spindle events in NREMS of WT and Ca$_V$3.3 KO mice. We followed a previously established thresholding approach in rat that successfully characterized spindles in both rodent and human (*Mölle et al., 2009*) complemented with additional criteria, as detailed in the Materials and methods (*Figure 5—figure supplement 1*). For the band-pass filtering, we were guided by the observation that in both S1 and S2 of the Ca$_V$3.3 KO animals, power was attenuated beyond the widely used sigma band of 10–15 Hz. Therefore, we chose 9–16 Hz to allow for the possible inclusion of comparatively slow and fast spindles.

We isolated 727–2289 events per WT mouse and area that showed the typical spindle-shaped, waxing-waning waveform (*Figure 5A*). Spindles in S1 and S2 showed large amplitudes that were comparable to that of the SO, whereas those in AC and PFC were less prominent (*Figure 5A*). In the Ca$_V$3.3 KO animals, the large spindle events were reduced in S1 and S2, but remained comparable in AC. There was also a reduction of event amplitude in PFC (*Figure 5B1*). Cumulative probability density curves showed a marked leftward shift of the amplitude distribution in S1 and S2, but not in AC and PFC (*Figure 5B2*). We also analyzed the intra-spindle frequencies, one of the major markers of spindle heterogeneity (*Figure 5C*). The frequency of detected events was distributed according to a Gaussian between 9 and 16 Hz, with a maximum around 10–12 Hz, yielding means of 11.6 ± 0.09 Hz for S1; 11.7 ± 0.05 Hz for S2; 11.2 ± 0.05 Hz for AC; 11.5 ± 0.03 Hz for PFC (*Figure 5C1*). All distributions showed a tail indicating a small proportion of events with a frequency >14 Hz (*Figure 5C2*). In the Ca$_V$3.3 KO animals, frequencies were specifically attenuated in S1 and S2, but remained comparable in AC and PFC. The greater activity of Ca$_V$3.3 channels in the sensory sectors of TRN thus correlated strongly with higher amplitudes and frequencies of individual spindles in S1 and S2.

## Ca$_V$3.3 channels ensure the temporal coordination of sleep spindles with the active state of the SO

Sleep spindles are temporally grouped by the active state of the SO, which reflects the strong role of corticothalamic volleys in recruiting thalamic circuits (*Contreras et al., 1996*). This grouping is key for the promotion of sleep-dependent memory consolidation (*Mölle et al., 2009*) and it may be

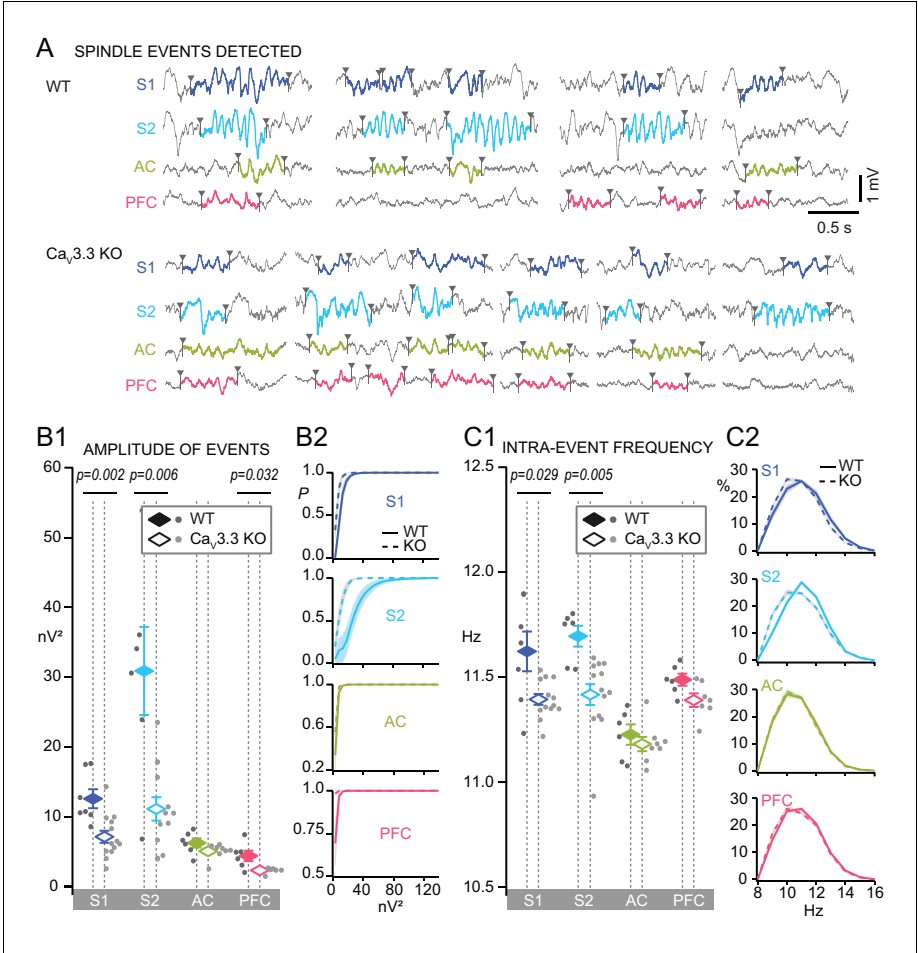

**Figure 5.** Discrete spindle events are amplified and accelerated in somatosensory areas by Ca$_V$3.3 channels. (**A**) Representative traces during NREMS for WT and Ca$_V$3.3 KO for S1, S2, AC and PFC, showing examples of algorithm-detected discrete spindle events (9–16 Hz, colored and bordered by arrowheads). (**B**) Mean amplitude of detected spindles quantified as mean power of the band-pass filtered signal (9–16 Hz). (**B1**) Mean spindle power levels across animals, values for individual animals shown by dots (dark gray for WT, light gray for Ca$_V$3.3 KO animals) and mean values ± S.E.M. by colored diamonds. Statistical significance was tested for each area separately using Mann-Whitney test, comparing WT (S1, n = 7; S2, n = 6; AC, n = 6; PFC, n = 6) and Ca$_V$3.3 KO (S1, n = 13; S2, n = 13; AC, n = 8; PFC, n = 7). p-values obtained were: for S1, p=0.002; for S2, p=0.006; for AC, p>0.05; for PFC, p=0.032. (**B2**) Cumulative probability distributions. (**C**) Same for intra-spindle frequencies. (**C1**) p-values obtained were: for S1, p=0.029; for S2, p=0.005; for AC and PFC, p>0.05. (**C2**) Probability distribution of spindle events according to their intra-spindle frequency.

DOI: https://doi.org/10.7554/eLife.39111.013

The following source data and figure supplement are available for figure 5:

**Source data 1.** Numerical data values and statistics underlying *Figure 5*.
DOI: https://doi.org/10.7554/eLife.39111.015

**Figure supplement 1.** Illustration of procedure for automated spindle detection Representative traces for the four areas, S1, S2, AC and PFC in a WT animal, showing for each (from top to bottom): raw trace, band-pass filtered 9–16 Hz, power of the filtered trace.
DOI: https://doi.org/10.7554/eLife.39111.014

disrupted in some cases of schizophrenia (*Manoach et al., 2016*). We determined the onset times of detected spindles with respect to the phase of the SO (*Figure 6A*) and confirmed such time-locking throughout all areas, with ~60% and 40% of detected spindles initiating during the cortical active and silent states (also named UP and DOWN states), respectively (*Figure 6B,C*). Remarkably, the same analysis in the Ca$_V$3.3 KO animals showed that the time-locking of sleep spindles to the SO

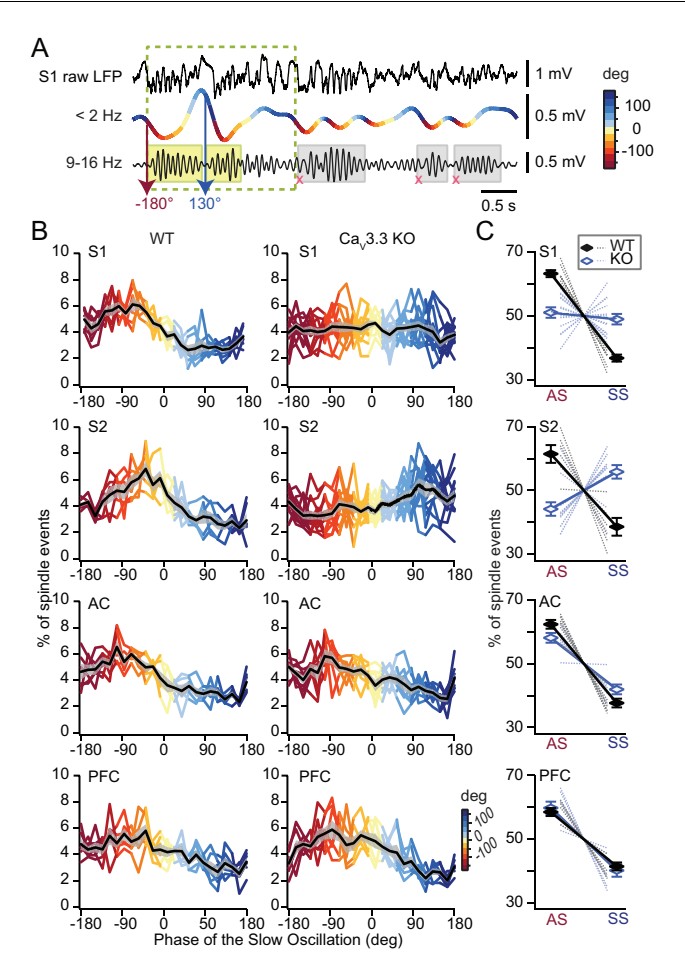

**Figure 6.** Phase-locking of spindles to the active state of the SO depends on Ca$_V$3.3 channels in somatosensory cortices. (A) Illustration of the method to determine the phase value of the SO at which a detected spindle event starts (yellow shaded rectangles). SO periods were determined based on (*Mölle et al., 2009*) and indicated as dashed rectangle. Spindles that did not coincide with detected SO were not included in this analysis (gray shaded rectangles). (B) Graphs presenting sleep spindle occurrence as a function of SO phase. Colored lines represent individual animals, black lines the mean ±S.E.M. (gray-shaded curve). Same animal numbers as in *Figure 5*. (C) Mean occurrence of spindle onsets for the active (AS, −180°, 0°) and the silent (SS, 0°, 180°) state of the SO. Statistical significance was tested for each area separately with respect to spindle occurrence as a function of AS and SS. All tests were paired *t*-test, except for S2 for which we used Wilcoxon signed rank-test. For WT, p-values obtained were: for S1, p=2×10$^{-5}$; for S2, p=0.031; for AC, p=3×10$^{-4}$; for PFC, p=0.001. For Ca$_V$3.3 KO, p-values obtained were: for S1, p=0.543; for S2, p=0.033; for AC, p=0.001; for PFC, p=0.003. Comparison between genotypes was done for the active state (AS) using unpaired *t*-test, except for S2 for which we used Mann-Whitney test. p-values obtained were: for S1, p=8.3×10$^{-6}$; for S2, p=5×10$^{-4}$; for AC, p=0.06; for PFC, p=0.59.
DOI: https://doi.org/10.7554/eLife.39111.016

The following source data is available for figure 6:

**Source data 1.** Numerical data values and statistics underlying *Figure 6*.
DOI: https://doi.org/10.7554/eLife.39111.017

was altered for S1 and S2 (*Figure 6B,C*), but not for AC and PFC. In both S1 and S2, the presence of spindles on the active state was diminished in favor of spindles in the silent state. In S1 from Ca$_V$3.3 KO animals, the occurrence of spindles was comparable between active and silent states, whereas events occurred preferentially in the silent state in S2. This difference in the coupling was not due to a less faithful detection of the smaller spindles in the Ca$_V$3.3 KO animals, because the analysis produced similar results when it was limited to WT spindles with amplitudes corresponding to those of

the Ca$_V$3.3 KO animals. Therefore, the presence of Ca$_V$3.3 ensures the timely occurrence of sleep spindles with respect to the cortical active state.

## Discussion

We identify a novel mechanism underlying local cortical correlates of NREMS. It arises through heterogeneity in thalamic circuits and correlates with variable oscillatory bursting propensity across TRN sectors. Strong Ca$_V$3.3-dependent bursting in the somatosensory TRN sector led to local NREMS with fast and large spindles coupled to the SO, whereas this was not the case for cortical areas corresponding to TRN sectors with weakly bursting cells. Intriguingly, high burst propensity was also coupled to a dual regulation of local NREMS, such that the SO and spindles could be suppressed in favor of delta waves. Sectorial attributes of TRN circuits thus emerge as a powerful source for local NREMS correlates. They additionally facilitate a tuning of NREMS between major spectral correlates, offering a candidate mechanism for local sleep modulation, possibly in response to use, experience and learning. A sectorial heterogeneity of TRN will also advance insight into a proposed common vulnerability of TRN circuitry for both sleep as well as wakefulness-driven selective attention in neuropsychiatric disorders (*Krol et al., 2018*).

The TRN has been traditionally regarded as a homogeneous sleep rhythm pacemaker (*Fuentealba and Steriade, 2005*; *Fogerson and Huguenard, 2016*) that underlies global thalamic inhibition (*Halassa and Acsády, 2016*). Although its molecular and functional heterogeneity is increasingly recognized, consequences for NREMS are so far poorly explored and the role of sector-specific TRN cell characteristics is unknown. Thus, it is not clear whether heterogeneous cellular discharge properties (*Contreras et al., 1992*; *Brunton and Charpak, 1997*; *Lee et al., 2007*; *Kimura et al., 2012*; *Higashikubo and Moore, 2018*) or variable expression of PV and somatostatin are important for local sleep (*Clemente-Perez et al., 2017*). Furthermore, sensory but not limbic TRN cells preferentially engage in sleep-related activity (*Halassa et al., 2014*), but it is not known whether they are cellularly distinct. Therefore, this study is the first to bring together TRN heterogeneities at both the *in vitro* and the *in vivo* levels using optogenetically assisted mapping of TRN sectors. We show that cellular heterogeneities align with identified TRN sectors, and TRN sectors with area-specific NREMS, thus establishing the TRN sector-specific cellular properties as a source for local and tunable NREMS. Based on a previous study demonstrating that PV-positive cells are stronger bursters than cells enriched in somatostatin (*Clemente-Perez et al., 2017*), we propose that PV-positive TRN cells are primary determinants of the unique sleep spindle properties in somatosensory cortex.

The focus on NREMS in specific cortical areas using LFP recordings, combined with genetic and chemogenetic manipulation and cellular analysis, reveals novel and surprising topographical aspects of mouse NREMS (*Terrier and Gottesmann, 1978*; *Kim et al., 2015*; *Fernandez et al., 2017*). One remarkable observation is that somatosensory cortices showed a clear sigma power 'shoulder' and strong, fast, Ca$_V$3.3-dependent spindle events. The widespread idea from EEG recordings that sleep spindle generation in mouse NREMS is weak (*Astori et al., 2011*) should thus be revised as we now show that full-fledged spindle activity is generated in local regions of the mouse brain. The highly focal synaptic organization of the barrel system, where TRN-thalamic, thalamocortical and cortico-thalamic projections between barreloid and barrel topographically match on a cell-to-cell basis (*Desîlets-Roy et al., 2002*; *Wimmer et al., 2010*), together with a high density of PV-positive, bursty cells in the TRN somatosensory sector (*Clemente-Perez et al., 2017*), are likely important anatomical substrates enabling strong local spindles. The tight alignment of thalamic unit and S1 LFP activity recorded simultaneously during spindles in urethane anaesthesia (*Rovó et al., 2014*) supports this interpretation. We also observed prominent sigma power and high-amplitude spindles in S2 that exceeded levels in S1. High reciprocal cortical connectivity between S1 and S2 (*Feldmeyer et al., 2013*), and a differential recruitment of first- and higher order thalamic nuclei in these two areas, could be some of the reasons behind this difference. In contrast to S1 and S2, no discernable sigma power shoulder was present in AC, and detected spindles showed no dependence on the Ca$_V$3.3 channel. We found a tendency for a decreased density of highly burst-prone cells in mouse, which possibly reduces the strength of spindle generation in auditory sectors. It also remains to be seen whether there exist functional and anatomical differences in the overall connectivity of each sensory TRN sector (*Crabtree, 1999*). Similarly, the spectral composition of NREMS in pre- and infralimbic

portions of the PFC showed no prominent sigma power and no overall dependence on $Ca_V3.3$ channels, consistent with a minor expression of $Ca_V3.3$ channels in the MD-innervated sector and with the presence of a majority of somatostatin-positive, weakly bursting cells in these areas of the TRN (*Clemente-Perez et al., 2017*). However, discrete spindles could be detected in mouse and they are well-described for rat PFC (*Siapas and Wilson, 1998*; *Peyrache et al., 2011*; *Maingret et al., 2016*). Discrete spindle events were accompanied by a phase entrainment of TRN discharge in sleeping rats, although mean firing rates seemed low (*Gardner et al., 2013*). There could thus be a spindle-generating circuitry independent of the powerful $Ca_V3.3$-dependent mechanisms in sensory TC loops, in which anterior and midline thalamic nuclei and hippocampus are involved. Such ideas remain to be tested also regarding the notion of distinct frontal spindles in both human (*Schabus et al., 2007*) and mouse (*Kim et al., 2015*).

Power in the SO band was consistently co-modulated with sleep spindle activity in both the $Ca_V3.3$ KO mouse and during chemogenetic TRN inhibition. This could be evidence in favor of a role of TRN in the generation of the SO (*Crunelli et al., 2015*). Our chemogenetic results generally support the idea that the TRN sustains low-frequency activity in cortico-thalamocortical loops, an effect which could possibly also result secondarily from its phasic recruitment by cortex to generate sleep spindles. Also, the increase of delta power could have de-emphasized the relative presence of SOs. Additionally, possible roles of $Ca_V3.3$ channels in cortically generated rhythms could arise from a subgroup of cortical interneurons (*Liu et al., 2011*).

The genetic removal of the $Ca_V3.3$ channel enhanced delta power in S1 and S2 at the expense of sleep spindles. Chemogenetic hyperpolarization reproduced this observation closely. Although the chemogenetic approach suppresses excitability in a manner that is not limited to bursting, the close correspondence with the observations in the $Ca_V3.3$-KO animals, in which only bursting but not tonic firing is impaired (*Astori et al., 2011*), strongly suggests that suppressed bursting, which is the common denominator of both experimental approaches, is the principal mechanism involved in the effects at the level of NREMS. While the reduction in sleep spindle activity under these conditions is expected (*Astori et al., 2011*), an enhancement and/or unmasking of delta wave-generating activity in thalamocortical circuits has now become apparent thanks to the locality of our recordings. It is reminiscent of previous observations of an opposite regulation of slow wave/delta and spindle power during NREMS (*Dijk et al., 1993*; *Steriade et al., 1993*; *Franken et al., 1998*), which has been explained based on the different membrane potential polarizations of TC cells involved in spindle or delta rhythm generation (*Nuñez et al., 1992*). Here, we identify TRN cell membrane potential polarization as a determinant for such regulation in somatosensory thalamocortical circuits during NREMS. Remarkably, the capability of TRN in generating either spindles or delta waves was also evident based on whether brief or prolonged optogenetic stimulation was applied (*Halassa et al., 2011*; *Lewis et al., 2015*). This underscores the power of TRN-dependent inhibition in controlling thalamocortical synchrony according to discharge patterns. We add to this the capability of a local, switchable tuning of NREMS spectral properties in somatosensory cortex. In agreement with pioneering work on delta waves, we propose that TRN hyperpolarization liberates TC cells from phasic hyperpolarization to engage in a clock-like rhythm at delta frequencies (*Steriade et al., 1993*). Results consistent with this interpretation were also obtained in animals doubly deficient in $Ca_V3$ channels (*Pellegrini et al., 2016*). Recently, it was also shown that optogenetic inhibition of anterior TRN cells may suppress rather than strengthen low-frequency EEG activity (*Herrera et al., 2016*), yet this effect occurred with a 10-s-long delay after acute inhibition of TRN cell discharge.

The powerful control of delta power by TRN-dependent mechanism could be relevant for the role of delta waves in the homeostatic regulation of NREMS. The increase in low-frequency power of NREMS over the 0.75–4 Hz power band, referred to as slow-wave activity, is the most widely used marker to quantify homeostatic sleep pressure (*Borbély and Tobler, 2011*). We find here that slow-wave activity contains a thalamically controlled component that can be rapidly and bidirectionally modulated through TRN membrane polarization. Such mechanisms could contribute to sleep-deprivation induced boosting of the high- but not the low-frequency component of slow-wave activity (*Achermann and Borbély, 1997*; *Huber et al., 2000*). Bidirectional regulation of slow-wave activity in local brain areas according to use dependence has also been described (*Kattler et al., 1994*; *Pigarev et al., 1997*; *Vyazovskiy et al., 2000*; *Miyamoto and Hensch, 2003*; *Huber et al., 2006*). More complex localized alterations in NREMS are observed following exposure to learning tasks that involve enhanced power in both the low-frequency range (SO and delta waves in the slow wave

activity frequency range of 0.5–4 Hz) and in the fast sleep spindle range (*Huber et al., 2004*). Learning tasks involving motor cortex increase the density of individual spindle events in a manner specifically restricted to motor cortex (*Johnson et al., 2012*) or augmented both slow wave and sleep spindle power in supplementary motor cortex (*Tamaki et al., 2013*). Therefore, cortical modules are capable of generating qualitatively different forms of local NREMS spontaneously and according to recent use and experience. Membrane potential polarization within TRN sectors could represent a powerful addition to previously proposed mechanisms that primarily imply changes in cortical synaptic strength (*Tononi and Cirelli, 2014*). Both ascending brainstem and basal forebrain (*McCormick and Bal, 1997*; *Beierlein, 2014*) as well as descending cortical inputs (*McCormick and von Krosigk, 1992*; *Zhang et al., 2012*) regulate TRN membrane potential and burst propensity. Whether differential neuromodulation of TRN sectors contributes to use- and experience-dependent sleep regulation remains an intriguing topic for further study.

Compromised sleep spindle generation is a promising read-out for neuropsychiatric disorders involving aberrant sensory percepts and attentional deficits, such as schizophrenia (*Manoach et al., 2016*). In large-scale genome-wide association studies, the gene encoding $Ca_V3.3$ channels ranks in the top list of candidate risk genes in schizophrenia, together with several genes that are highly enriched in TRN and implied in repetitive burst discharge (*Krol et al., 2018*). Based on our data, we propose that wake-related deficits in some of these patients may show a specificity for certain sensory modalities that co-vary with local deficits in sleep spindles and their coupling to the SO. This could help to further refine the classification and diagnosis of these complex disorders.

# Materials and methods

## Key resources table

| Reagent type (species) or resource | Designation | Source or reference | Identifiers | Additional information |
|---|---|---|---|---|
| Genetic reagent (*M. musculus*) | $Ca_V3.3$ KO | PMID: 21808016 | MGI:5637591 | generated by Dr. H. Prosser, then at GSK |
| Genetic reagent (*M. musculus*) | VGAT-Ires-Cre | PMID: 21745644 | MGI:5141270 | generated by Dr. B. Lowell, Harvard |
| Recombinant DNA reagent | AAV1-hSyn-ChR2(H134R)_eYFP-WPRE-hGH | Penn Vector Core | 26973P | |
| Recombinant DNA reagent | AAV8-hSyn-DIO-hM4D(Gi)_mCherry | UNC Vector Core | N/A | |
| Recombinant DNA reagent | ssAAV8/2-hSyn1-dlox-HA_hM4D(Gi)_IRES_mCitrine-dlox-WPRE-hGHp(A) | Zurich viral vector repository | v93-8 | |
| Antibody | mouse anti-PV RRID:AB_10000343 | Swant | PV 235 | Dilution 1/4000 |
| Antibody | goat anti-mouse CY5 RRID:AB_2338713 | Jackson ImmunoResearch | 115-175-146 | Dilution 1/500 |
| Peptide, recombinant protein | streptavidin coupled with Alexa Fluor 594 RRID:AB_2337250 | Jackson ImmunoResearch | 016-580-084 | Dilution 1/8000 |
| Chemical compound, drug | CNO | Tocris | 6329 | |
| Software, algorithm | Neuroexplorer | Plexon | | |

*Continued on next page*

*Continued*

| Reagent type (species) or resource | Designation | Source or reference | Identifiers | Additional information |
|---|---|---|---|---|
| Software, algorithm | Intan RHD2000 recording system with Matlab toolbox 1.2.2 | IntanTeck | | |
| Software, algorithm | PClamp10.2 | Molecular Devices | | |
| Software, algorithm | Igor Pro 7 | WaveMetrics | | |
| Software, algorithm | Matlab 2018 a | MathWorks | | |
| Software, algorithm | R 3.5.1 | R Core Team | | |

## Animal handling

Mice from the C57BL/6J line (also referred to as wild-type, WT), the $Ca_V3.3$ KO line and the VGAT-Ires-Cre line (Jackson Labs, generated by Dr. B. Lowell, Beth Israel Deaconess Medical Center, Harvard) (*Vong et al., 2011*) were bred on a C57BL/6J background and housed in a temperature- and humidity-controlled animal house with a 12 hr/12 hr light-dark cycle (lights on at 9 am). Food and water were available *ad libitum*. For viral injections, 3- to 4-week-old mice of either sex were transferred to a P2 safety level housing room with identical conditions 1 day prior to injection. Then, for *in vitro* experimentation, animals were transferred 3 to 4 weeks later to a housing room with identical conditions, 3–5 days prior to sacrifice. For *in vivo* experimentation, animals were brought to the recording room at least one week prior to experimentation. All experimental procedures complied with the Swiss National Institutional Guidelines on Animal Experimentation and were approved by the Swiss Cantonal Veterinary Office Committee for Animal Experimentation.

## Viral injections

Mice 3- to 4-week-old were anaesthetized using Ketamine-Xylazine (83 and 3.5 mg/kg, respectively). Mice were placed on a heating blanket to maintain the body temperature at 37°C. An initial dose of analgesic was administrated at the beginning of the surgery (Carprofen i.p. 5 mg/kg). The animal was head-fixed on a stereotactic apparatus equipped with a head adaptor for young animals (Stoelting 51925, Wood Dale, IL). A small incision was made on the skin and the bone exposed at the desired injection site. Viruses were injected with a thin glass pipette (5-000-1001-X, Drummond Scientific, Broomall, PA) pulled on a vertical puller (Narishige, Tokyo, Japan). WT and $Ca_V3.3$ KO mice were injected bilaterally with a virus encoding ChR2-EYFP (500 nl of AAV1-hSyn-ChR2(H134R)_eYFP-WPRE-hGH, $10^{12}$ GC, ~100–200 nl/min) for one of the following sites (in stereotaxic coordinates, relative to bregma: anteroposterior, lateral, depth from surface): S1 (-1.7, ±3.1, -0.8), AC (-2.5, ±4, -1.1), MD (-1.7, ±0.4, -3.2). VGAT-Ires-Cre mice were injected bilaterally with a virus encoding DREADD-mCherry (500 nl of AAV8-hSyn-DIO-hM4D(Gi)_mCherry, $6.4x10^{12}$ GC), or DREADD-IRES-mCitrine (500 nl of ssAAV8/2-hSyn1-dlox-HA_hM4D(Gi)_IRES_mCitrine-dlox-WPRE-hGHp(A), $3.1x10^{12}$ GC) or a control AAV8 encoding a DREADD-unrelated construct (500 nl of AAV8-hSyn-FLEX-Jaws_KGC_GFP_ER2, $3.2x10^{12}$ GC) in the sensory sector of the TRN (-1.7, ±2.25, -2.9).

## *In vitro* electrophysiological recordings

Adult WT, $Ca_V3.3$ KO and VGAT-Ires-Cre mice (3–4 weeks post viral injection), 7- to 9-week-old, were briefly anaesthetized with isoflurane and their brains quickly extracted. Acute 300-μm-thick coronal brain slices were prepared in ice-cold oxygenated sucrose solution (which contained in mM: NaCl 66, KCl 2.5, $NaH_2PO_4$ 1.25, $NaHCO_3$ 26, *D*-saccharose 105, *D*-glucose 27, *L(+)*-ascorbic acid 1.7, $CaCl_2$ 0.5 and $MgCl_2$ 7), using a sliding vibratome (Histocom, Zug, Switzerland). Slices were kept for 30 min in a recovery solution at 35°C (in mM: NaCl 131, KCl 2.5, $NaH_2PO_4$ 1.25, $NaHCO_3$ 26, *D*-glucose 20, *L(+)*-ascorbic acid 1.7, $CaCl_2$ 2, $MgCl_2$ 1.2, *myo*-inositol 3, pyruvate 2) before being transferred to room temperature for at least 30 more min before starting the recording.

Recording glass pipettes were pulled from borosilicate glass (TW150F-4) (World Precision Instruments, Sarasota, FL) with a DMZ horizontal puller (Zeitz Instruments, Martinsried, Germany) to a final resistance of 2–4 MΩ. Pipettes were filled with a $K^+$-based intracellular solution that contained in mM: KGluconate 140, Hepes 10, KCl 10, EGTA 0.1, phosphocreatine 10, Mg-ATP 4, Na-GTP 0.4, pH 7.3, 290–305 mOsm, supplemented with ~2 mg/ml of neurobiotin (Vector Labs, Servion, Switzerland). Slices were placed in the recording chamber of an upright microscope (Olympus BX50WI, Volketswil, Switzerland) and continuously superfused at room temperature with oxygenated ACSF containing in mM: NaCl 131, KCl 2.5, $NaH_2PO_4$ 1.25, $NaHCO_3$ 26, D-glucose 20, L(+)-ascorbic acid 1.7, $CaCl_2$ 2 and $MgCl_2$ 1.2, picrotoxin 0.1, glycine 0.01. Cells were visualized with differential interference contrast optics and 10X and 40X immersion objectives. Infrared images were acquired with an iXon Camera X2481 (Andor, Belfast, Northern Ireland). Signals were amplified using a Multiclamp 700B amplifier, digitized via a Digidata1322A and sampled at 10 kHz with Clampex10.2 (Molecular Devices, San José, CA). Immediately after gaining whole-cell access, cell capacitance $C_m$ was measured in voltage-clamp at −60 mV through applying 500 ms-long, 20 mV hyperpolarizing steps (5 steps/cell). Whole-field blue LED (Cairn Res, Faversham, UK) stimulation (455 nm, duration: 0.1 to 1 ms, maximal light intensity 0.16 mW/mm$^2$) in voltage-clamp (−60 mV) was used to assess the connectivity of TRN neurons through fibers arising from the previously injected area (S1, AC or MD). Once identified, squared somatic current injections (−50 to −300 pA for 500 ms, 4 injections/cell and membrane potential) hyperpolarized neurons below −100 mV from membrane potentials between −90 and −50 mV (corrected for a liquid junction potential of 10 mV) and induced repetitive burst discharge in TRN neurons. For comparison between neurons from different TRN sectors, the response to this current injection was also used to assess cellular input resistance $R_i$. For CNO application, a stable baseline of at least 2 min was recorded before bath application of water-soluble CNO (10 μM, Tocris, Bristol, UK) for at least 2 min until its hyperpolarizing effect reached a plateau. Washout of CNO did not reverse the hyperpolarizing effect for up to 10 min of washout. Two VPM and two TRN neurons outside the visually identified fluorescent site of injection were used as control for CNO's effect on membrane potential.

Cell parameters were calculated on Clampfit v.10.2. and IgorPro Wavemetrics (Lake Oswego, OR). Input and access resistances were evaluated all along the recordings. Neurons presenting a variation of the access resistance >20% or a holding current at −60 mV < −150 pA were excluded from analysis. Light-evoked excitatory postsynaptic current (EPSC) amplitudes were measured in traces presenting monophasic synaptic events that occurred at fixed latency after LED onset and that were not contaminated by asynchronous release. In the subset of neurons that underwent paired-pulse stimulation, traces containing spontaneous activity between the two LED stimulations were discarded. Paired-pulse ratios are expressed as percentage of the second over the first EPSC amplitude (6 paired-stimuli/cell). The bursting of TRN neurons was assessed by counting the number of $Ca^{2+}$ spikes following a 500 ms hyperpolarization below −100 mV. $Ca^{2+}$ spikes were counted as bursts if they generated triangular-shaped membrane depolarizations followed by a clear afterhyperpolarization, regardless of the presence of high-frequency action potentials on top of the $Ca^{2+}$ spikes.

### In vivo multi-site electrophysiological recordings

Surgery was performed as recently described (*Lecci et al., 2017*). Briefly, animals were subjected to gas anaesthesia (isoflurane supplemented with a mixture $O_2$ and $N_2O$) and small craniotomies (0.3–0.5 mm) were performed at the location for implantation of high-impedance fine tungsten LFP microelectrodes (10–12 MΩ, 75 μm shaft diameter, FHC, Bowdoin, ME) at the following sites (in stereotaxic coordinates, relative to bregma: anteroposterior, lateral, depth from surface): sensory regions S1 (−1.7, 3.0, -1.0), S2 (−0.7, 4.2, -1.1), and AC (−2.5, 4.0, -1.1), limbic areas of PFC (+1.8, 0.3, -1.85). Implantations were guided through calculating the corresponding interaural coordinates. As a neutral reference for LFP electrodes, a silver wire (Harvard Apparatus, Holliston, MA) was inserted in the bone over lateral portions of the cerebellum. On the contralateral skull site, two conventional gold-coated low-impedance electrodes were implanted over the *dura mater* through frontal and parietal bones for differential surface EEG recordings. Two gold pellets inserted into the muscles of the neck served as EMG electrodes. Multi-site recordings were carried out in head-restrained conditions, for which a light-weight metal head-post (Bourgeois Mécanique SAS, Lyon, France) was glued and cemented onto the midline skull in order to perform painless head-fixed recording sessions (*Fernandez et al., 2017*; *Lecci et al., 2017*). Carprofen (5 mg/kg, i.p.) and

paracetamol (2 mg/mL, drinking water) were provided during the pre- and post-operative periods. Mice were gently and gradually habituated to a custom-made head-fixation system (Bourgeois Mécanique SAS, Lyon, France) by increasing the amount of time spent in head fixation daily from 10 to 30 min to 2–3 hr/day. Mice sat within a cardboard roll such that only the head protruded. Occasionally, a heating pad was placed underneath the cardboard. After each head-restrained period, mice were rewarded with *ad libitum* drops of sweetened water. Mice typically started sleeping spontaneously after 7–14 d of habituation, generating periods of both NREMS and REMS. LFP and EEG/ EMG signals were amplified 1000x through a 16-channel Multiple Acquisition Processor System (Plexon Inc., Dallas, TX), high- and low-pass filtered at 0.8 and 300 Hz, respectively, and digitized at 1 kHz. For multi-site recordings, 5 WT and 6 Ca$_V$3.3 KO animals had all four recording sites histologically confirmed, 1 WT (S1, S2, PFC) and 3 Ca$_V$3.3 KO (2x S1, S2, AC; 1x S1, S2, PFC) animals had three confirmed recording sites, 3 WT (1x S1, AC; 2x S1, S2) and 4 Ca$_V$3.3 KO (S1, S2) animals had two confirmed sites. Four animals were excluded because recording sites could not be identified or because brain appearance was not satisfying (ventricle dilatation or damaged brain slice).

## *In vivo* chemogenetics

After 1 week of recovery from viral injection, bilateral S1 LFP electrodes, and EEG/EMG electrodes were implanted as described above, followed by a week of recovery. After 4 days of habituation to the tethering cable, baseline activity was recorded in freely moving conditions during 3 to 4 days. 3 weeks after viral injections, intra-peritoneal injection of CNO (water-soluble diluted in NaCl 0.9%, dosis 1 mg/kg, Ref. 6329, Tocris, Bristol, UK) or NaCl 0.9% was performed at Zeitgeber time ZT2 in a random cross-over design, with the experimenter blind to the injection. Recordings under each condition took place on 4–5 successive days, with 2–3 CNO and 1–2 NaCl injections per animal. Signals were acquired at 1 kHz with an Intan digital RHD2132 amplifier board and a RHD2000 USB Interface board, with a high-pass filter set at 0.8 Hz (Intan Technologies, Los Angeles, CA). Data were acquired in Matlab using the RHD2000 Matlab toolbox and customized display software in Matlab. To assess the time course of CNO action, 3 DREADD-mCherry and 3 AAV8-control animals were recorded starting at ZT0. Based on these dynamics, we chose a window of 45 min (starting 20 min after the injection) for comparison of CNO and NaCl effects (*Figure 4—figure supplement 1B, D*). The average power spectrum per animal was calculated as mean between recording sites and repetition-days for the two conditions (CNO or NaCl). Spectral analysis of the signals was performed as described in the data analysis section. For chemogenetic recordings, a total of 3 DREADD-mCherry and 2 control AAV8 mice had bilateral S1 recording sites histologically confirmed, 2 DREADD-mCherry and 1 control AAV8 had one confirmed S1 recording sites.

## Histology and immunofluorescent labeling

After completion of patch-clamp recordings *in vitro*, slices were post-fixed in paraformaldehyde (4%) for >24 hr. An immunostaining on free-floating sections was used to outline PV-positive (PV+) neurons in the TRN and to recover neurobiotin-filled neurons. To ensure proper staining of PV+ neurons, a 5-day incubation at 4°C of the primary antibody (mouse anti-PV, 1/4000, Swant Inc., Marly, Switzerland) in 1% Triton was required. The secondary antibody (goat anti-mouse CY5, 1/500, Jackson ImmunoResearch, Ely, Nevada) and Streptavidin (coupled with Alexa Fluor 594, 1/8000, Jackson ImmunoResearch) were incubated at 4°C for 24 hr. Sections were observed with an Axiovision Imager Z1 (Carl Zeiss) microscope equipped with an AxioCam MRc5 camera. Objectives were EC-Plan Neofluar 2.5x/0.075 ∞/0.17, 5x/0.16 ∞/0.17. The AxioVision Rel. 4.7 and Adobe Photoshop CS5 software were used to merge micrographs from the different channels. The nomenclature of the location of TRN sectors in *Figure 1* follows the descriptions established by *Pinault and Deschênes (1998)*.

After completion of *in vivo* recordings (multi-site or chemogenetics recordings), recording sites were marked through electro-coagulation (50 µA, 8–10 s) during deep pentobarbital anaesthesia (80 mg/kg) before transcardiac perfusion (4% paraformaldehyde in 0.1 M phosphate buffer). After >24 hr post-fixation, 100 µm coronal brain sections were cut and imaged to confirm electrocoagulation sites of LFP implantation or fluorescent expression of viral injection site.

### *In vivo* data analysis

Data were analyzed using IgorPro (Wavemetrics, v7, Lake Oswego, OR), MatLab (MathWorks) and Excel (Microsoft).

### Scoring of vigilance states

Sleep and wake episodes were detected manually according to standard scoring procedures (*Fernandez et al., 2017*). Briefly, wakefulness was accompanied by large or tonic EMG signal (active and quiet wakefulness), EEG of low-voltage and exhibiting fast oscillatory components, such as theta and gamma oscillations. Drowsiness period between wakefulness and NREMS were discarded, as well as intermediate sleep periods between NREMS and REMS. Only consolidated clear episodes of NREMS were selected for the analysis: high amplitude voltage and slow EEG components, such as periods of slow oscillation (<1.5 Hz), delta waves (1.5–4 Hz) and spindles (10–15 Hz). REMS was clearly distinguishable with reduced EMG activity (atonia) and predominant theta (~6–10 Hz) on the low-amplitude EEG. For the chemogenetic data analysis, NREMS was scored in 4 s epochs with the same criteria and only consolidated NREMS (>20 s) were included in further analysis. All scorings were done with custom-made software prepared in Igor and Matlab.

### Spectral analysis of the signals

Power spectra were computed on raw signals using a squared Fast Fourier Transform (FFT) on 4 s windows after offset correction (mean substracted for each window). Each mean power spectrum per mouse and per channel was normalized by its sum between 0 and 35 Hz and expressed in per-centage to compare between animals. The total power per frequency band of interest, SO 0.5–1.5 Hz, delta 1.5–4 Hz and sigma 10–15 Hz, was measured using the integral of the normalized power spectrum in-between frequency band borders.

### Dynamics of delta time course (Chemogenetics)

The dynamics of delta activity were extracted from the area under the power spectrum in the delta band (1.5–4 Hz) for each 4 s epoch of consolidated NREMS (>20 s continuous bouts). The time-series were calculated in quantiles of identical amounts of NREMS (12 for baseline and 36 for post-injections) and normalized by the mean of the first two baseline bins.

### Spindle event detection

The square-power of the filtered enlarged sigma band (9–16 Hz, Finite Impulse Response) calculated separately for the recordings from each brain area was used to detect onset and offset of spindles. We applied a threshold of [1.5 x the S.D. + 1 x the mean] of the sigma power, and detected all events above this threshold that lasted at least 3 cycles (*Figure 5—figure supplement 1A*). The onset and offset times of a spindle event were extended to the closest cycle at 0 crossing before and after the threshold. Events that were overlapping or that were separated by <10 ms were fused. Events that were at the beginning time point or last time point of a NREMS bout were discarded. Amplitudes of spindles were computed from the average amplitude of the sigma power between onset and offset time and averaged per mouse. Frequency of spindles were determined by extracting the peak frequency from the magnitude FFT on each spindle event (86.3% of events had a distinguishable frequency peak and were included), followed by calculating the mean intra-frequency per mouse. Two WT mice with S1 and S2 recording sites were omitted due to an unusually reduced signal amplitude that prevented comparison to other mice (note that no normalization was applied for spindle detection).

### Slow oscillation detection

Periods of clear and visible SOs were detected based on (*Mölle et al., 2009*). Briefly, each signal was 2 Hz low-pass filtered, minimum (y1) and maximum (y2) were detected, as well as the corresponding time point (x1 and x2). The mean minima (Y1) and the mean maxima (Y2) as well as their difference (Y2-Y1) were calculated. Constraints for selecting periods of SOs were: (1) times between x1 and x2 were comprised between 0.5 s and 2 s, (2) if y1 was lower than 2/3 of the mean Y1, (3) y2-y1 was at least 2/3 of Y2-Y1. These constraints allowed to select the largest SOs periods. Only events

that had a time overlap ≥95% to a SO were selected for the phase-locking analysis. Angle phase values of the SO at each spindle event onset time detected were extracted using a Hilbert transform.

## Statistics

*In vitro*. Statistical analysis was done using R programming language (2.15.0, R Core Team) [The R Development Core Team, The R Foundation for Statistical Computing (www.r-project.org/foundation), 2007]. The normality of the data sets was assessed using Shapiro-Wilk normality test. Comparisons between paired conditions (amplitude of 1st versus 2nd EPSCs during paired-stimulation and effect of CNO on membrane potential) were done using Wilcoxon signed rank-test and paired Student's *t*-test, respectively. Comparisons between unpaired conditions in non-normally distributed datasets (passive cellular properties, sector effect on repetitive bursting in WT, sector effect on repetitive bursting in $Ca_V3.3$ KO, genotype effect on repetitive bursting) were done using Mann-Whitney or Kruskal-Wallis H tests. Comparisons between unpaired conditions in normally distributed datasets (sector and genotype effect onto PPR and CNO effect onto WT vs $Ca_V3.3$ KO) were done using unpaired Student's *t*-tests or two-way ANOVAs. Bonferroni correction was applied whenever more than two comparisons were done for the same data set. The proportion of repetitive bursting neurons in the different TRN sectors was compared using Chi-square test for independence followed by a pairwise proportion test with Holm's adjustment method. Exact significant p-values are indicated.

*In vivo*. Statistical analysis was done using IgorPro, R programming language and Matlab. The normality of the data sets was assessed using Shapiro-Wilk normality test. Comparisons between genotypes per site of recording were done using Student's *t*-test (parametric data set) or Mann-Whitney (non-parametric unpaired data set). Comparisons between paired conditions (CNO versus NaCl, or SO Active state versus Silent state) were done using paired Student's *t*-test (parametric data set) or Wilcoxon signed-rank test (non-parametric paired data). Bonferroni correction was applied when more than two comparisons were done for the same data set.

## Acknowledgements

We are grateful to all lab members for critical input at all stages of this manuscript. In particular, Sandro Lecci was heavily involved in supporting experimentation and in data analysis discussions. The excellent animal care and support by the Team of our Animalerie, headed by Alain Gnecchi and Michelle Blom, and the expert veterinary advice and support of Drs. Gisèle Ferrand and Laure Sériot are highly appreciated. We thank Christiane Devenoges for excellent technical support on histology and genotyping. We are indebted to Antoine Adamantidis, Simone Astori, Paul Franken, Manuel Mameli and Mehdi Tafti for many constructive discussions and experimental input. Simone Astori, Sylvain Crochet, Sandro Lecci and Francesca Siclari provided critical comments on prefinal versions of the manuscript. Many thanks to Christian Lüscher and Manuel Mameli for making the VGAT-Ires-Cre animals available to us. This study was supported by the Swiss National Science Foundation, the Fondation Pro-Femmes, and Etat de Vaud.

## Additional information

### Funding

| Funder | Grant reference number | Author |
| --- | --- | --- |
| Swiss National Science Foundation | 31003A_166318 | Laura MJ Fernandez<br>Gil Vantomme<br>Alejandro Osorio-Forero<br>Romain Cardis<br>Elidie Béard<br>Anita Lüthi |
| État de Vaud | | Laura MJ Fernandez<br>Gil Vantomme<br>Alejandro Osorio-Forero<br>Romain Cardis<br>Elidie Béard<br>Anita Lüthi |

| FBM Poste de soutien à un congé parental | Laura MJ Fernandez |
|---|---|

The funders had no role in study design, data collection and interpretation, or the decision to submit the work for publication.

## Author contributions
Laura MJ Fernandez, Conceptualization, Data curation, Formal analysis, Supervision, Validation, Investigation, Visualization, Methodology, Writing—original draft, Project administration, Writing—review and editing; Gil Vantomme, Alejandro Osorio-Forero, Data curation, Formal analysis, Validation, Investigation, Visualization, Methodology, Writing—original draft, Writing—review and editing; Romain Cardis, Resources, Software, Formal analysis, Visualization, Methodology, Writing—review and editing; Elidie Béard, Data curation, Formal analysis, Validation, Investigation, Methodology, Writing—review and editing; Anita Lüthi, Conceptualization, Data curation, Supervision, Funding acquisition, Validation, Visualization, Writing—original draft, Project administration, Writing—review and editing

## Author ORCIDs
Laura MJ Fernandez http://orcid.org/0000-0002-7942-3369
Gil Vantomme http://orcid.org/0000-0002-7441-0737
Alejandro Osorio-Forero http://orcid.org/0000-0003-4341-4206
Anita Lüthi https://orcid.org/0000-0002-4954-4143

## Ethics
Animal experimentation: All experimental procedures complied with the Swiss National Institutional Guidelines on Animal Experimentation (Swiss Federal Act on Animal Protection, LPA 2005) and were approved by the Swiss Cantonal Veterinary Office Committee for Animal Experimentation. All experiments were carried out in accordance with approved protocols by the Swiss Cantonal Veterinary Office Committee for in vitro experimentation on mice (reference VD2062) and for in vivo experimentation on mice (references VD2387 and VD2401).

## Decision letter and Author response
Decision letter https://doi.org/10.7554/eLife.39111.022
Author response https://doi.org/10.7554/eLife.39111.023

# Additional files

## Supplementary files
• Transparent reporting form
DOI: https://doi.org/10.7554/eLife.39111.018

## Data availability
All data generated or analysed during this study are included in the manuscript and supporting files.

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
