## [Decision Letter]

Thank you for submitting your article "Thalamic reticular control of local sleep in mouse sensory cortex" for consideration by *eLife*. Your article has been reviewed by three peer reviewers, one of whom is a member of our Board of Reviewing Editors, and the evaluation has been overseen by Andrew King as the Senior Editor. The following individuals involved in review of your submission have agreed to reveal their identity: Jeanne Paz (Reviewer #2); Christopher I Moore (Reviewer #3).

The reviewers have discussed the reviews with one another and the Reviewing Editor has drafted this decision to help you prepare a revised submission.

Summary:

This study by Fernandez et al. identifies the TRN as a source for local, sensory-cortex-specific non-rapid-eye-movement sleep (NREMS) in mouse. The writing is clear, and the methodology and results are for the most part (see below), very strong. The authors put forward a clear case that local field potential recordings from different cortical regions exhibit distinct NREMS-related activity patterns, and link these to burst firing in associated areas of the TRN. In brain slices, they confirm that parts of TRN associated with sensory or non-sensory cortex show different bursting activity, and that this distinction is lost in CaV3.3 knockout mice. The authors conclude that the CaV3.3 channel endows higher bursting capacity to sensory over non-sensory TRN cells. Sensory cortices (especially S1) have more σ band activity and spindles compared to non-sensory (PFC), while reducing bursting by knocking out CaV3.3 or inhibiting TRN via DREADDS is sufficient to interfere with normal bursting, and shifts relative signal power away from σ and toward δ.

Essential revisions:

1) CNO induced hyperpolarization does not simply reduce TRN bursting. More likely it silences ALL TRN output, both bursting and non-bursting. Along these lines, CNO induces a significant increase in membrane conductance leading to reduced hyperpolarization from negative conditioning current pulses – this resultant reduced hyperpolarization would reduce the likelihood of a rebound burst and should be controlled for by increasing the current step.

2) Why were cortical injections chosen for sensory (S1, A1) TRN projections, but thalamic (MD) for "non"-sensory TRN projections? This is not very well rationalized.

3) The Discussion is quite long and speculative, arguably going beyond what is warranted from the specific results in this study.

4) In the end, there is somewhat limited information provided here regarding TRN heterogeneity beyond what is already known. Lee et al. and Clemente et al. have already documented similar differences in TRN bursting. What is new here is the spectral analysis of mPFC power during sleep and the effects of loss of Cav3.3 on this. Accordingly, the novelty of these experimental findings needs to be very clearly described in the Discussion section, by not only citing the previous papers but by being very explicit about what is novel here compared with previous studies.

5) Spindle power is mainly seen in S2. The reasons for this are not very well developed in the manuscript.

6) To determine if the alterations in NREMS of CaV3.3-/- animals were caused by suppressed TRN burst discharge the authors used chemogenetic inhibition of TRN cells and found that they could reproduce the switch from spindle- to δ-enriched sleep. The authors conclude that decreased TRN bursting produces an enrichment of δ power at the expense of σ and SO power in NREMS. How do the authors explain the enhanced δ activity when inhibiting TRN cells? Lewis et al., 2015 had showed that tonic optical activation of TRN enhances δ power in the cortex which seems inconsistent with the findings described in this study. How do the authors explain this discrepancy?

7) Given the previous findings of TRN heterogeneity (Clemente, Halassa), the authors should close the loop, and determine whether the diversity in CaV 3.3 behavior they observe is actually related to the highly parallel diversity observed between PV and SOM cell types in the Clemente et al. work. There is a mixing of PV and other cell types in the recipient zones the authors identified: Slice recordings, for example, from a PV-tagged animal, and comparison of these cells in the SI pathway with all others, would likely be enough, though SOM tagged recordings would also be informative.

Minor point:

The authors throw up their hands a little bit in the Discussion about how their algorithm might have impacted their findings. The authors should determine whether their algorithm impacts aspects of the behavior they observe. Specifically, they could sub-sample the spindles identified in the WT to be only from the same amplitude distribution as the Ca_V_3.3^-/-^ mice, and then see whether they still observe phase locking to the SO. This analysis should take just a few hours - they just need to limit their already conducted analyses to a smaller part of the data set - and would let them know much better if perhaps there is a fundamental mechanism at play.

---

## [Author Response]

Essential revisions:1) CNO induced hyperpolarization does not simply reduce TRN bursting. More likely it silences ALL TRN output, both bursting and non-bursting. Along these lines, CNO induces a significant increase in membrane conductance leading to reduced hyperpolarization from negative conditioning current pulses – this resultant reduced hyperpolarization would reduce the likelihood of a rebound burst and should be controlled for by increasing the current step.

As suggested by the reviewers, we have performed additional experiments during which we compensate for the increased membrane conductance induced by CNO. We additionally show now that counteracting CNO-induced hyperpolarization through direct current injections re-instates rebound bursting. We thus conclude that CNO thus does not block bursting directly but instead brings cells below threshold for rebound burst discharge. The corresponding Figure4A, B are now replaced with a new representative recording and new mean data and the corresponding text is entered in the first paragraph of the subsection “Chemogenetic inhibition of TRN cells reproduces the switch from spindle- to delta enriched sleep” and in the legend to Figure 4. The methodological information has been complemented on in the first paragraph of the subsection “In vitro electrophysiological recordings”.

2) Why were cortical injections chosen for sensory (S1, A1) TRN projections, but thalamic (MD) for "non"-sensory TRN projections? This is not very well rationalized.

The strategy to target a non-sensory area of the TRN was based on available classic literature that points to the MD as a major associative thalamic nucleus interacting with non-sensory TRN portions (see e.g. Pinault and Deschênes, 1998; Mitchell, 2016). Moreover, MD-prefrontal connections form important reciprocal thalamocortical loops (Delevich et al., 2015; Collins et al., 2018). In contrast, the literature on the anatomy of prefrontal-TRN synaptic projections is scarce in rodent, and the relationship between the multiple corticothalamic projections into the mouse MD (Mátyás et al., 2014) and their innervation of TRN is not known. Therefore, the identification of non-sensory TRN sectors was based on targeting the clearly identifiable MD. Based on the reviewer’s note, we have now further rationalized this point in the first paragraph of the subsection “TRN cell burst discharge propensity in acute slices varies across sensory and non-sensory sectors”.

3) The Discussion is quite long and speculative, arguably going beyond what is warranted from the specific results in this study.

We have accommodated this important concern mainly through two changes. First, we have carried out additional control analyses for the quality of our spindle detections. This has allowed us to remove the corresponding Discussion paragraph.

Second, we have further alleviated the Discussion through removing the comparison between species, which seems to us the most speculative part.

4) In the end, there is somewhat limited information provided here regarding TRN heterogeneity beyond what is already known. Lee et al. and Clemente et al. have already documented similar differences in TRN bursting. What is new here is the spectral analysis of mPFC power during sleep and the effects of loss of Cav3.3 on this. Accordingly, the novelty of these experimental findings needs to be very clearly described in the Discussion section, by not only citing the previous papers but by being very explicit about what is novel here compared with previous studies.

We realize that we need to more clearly state that this is the first study that identifies heterogeneities of the TRN based on the functional identification of cells through optogenetically assisted circuit mapping. The point here is that we worked with an a priori functional specification of thalamocortical circuits in which cellular properties were then studied. This also allowed to make a strong case for a subcortical determinant of local sleep properties, the TRN. To make this point more clear, we have reworded our statement in the second paragraph of the Discussion.

5) Spindle power is mainly seen in S2. The reasons for this are not very well developed in the manuscript.

We agree with the reviewers that spindle power is very pronounced in S2, the secondary somatosensory cortex known to be involved in sensory discrimination. We understand the reviewer’s comments as asking for the possible mechanistic reasons underlying this observation. A comprehensive answer would require additional studies to define the sources of sleep spindles in S2. For example, opto- or chemogenetic inhibition of S1 would help decide whether the well-described strong reciprocal wiring with the barrel cortex in S1 (Feldmeyer et al., 2013) is implied in amplifying spindles in S2. Furthermore, there is the possibility of a differential involvement of first- and higher-order thalamic nuclei in sleep spindle generation in S1 compared to S2. This question would also be addressable using opto- or chemogenetic interference. These possibilities remain open to experimentation and could shed light on what are probably multiple origins of cortical spindles. We allude to these possibilities through adding an additional sentence to the third paragraph of the Discussion.*6) To determine if the alterations in NREMS of CaV3.3-/- animals were caused by suppressed TRN burst discharge the authors used chemogenetic inhibition of TRN cells and found that they could reproduce the switch from spindle- to δ-enriched sleep. The authors conclude that decreased TRN bursting produces an enrichment of δ power at the expense of σ and SO power in NREMS. How do the authors explain the enhanced δ activity when inhibiting TRN cells? Lewis et al., 2015 had showed that tonic optical activation of TRN enhances δ power in the cortex which seems inconsistent with the findings described in this study. How do the authors explain this discrepancy?*

The reviewers note that a previously published study by Lewis et al. seems at odds with some of our observations. However, several technical and analytical points warrant attention that make us feel this conclusion is premature.

This study is mostly concerned with awake mice, only Figure 4D refers to sleeping mice. Full power spectra are not shown and power values are calculated by pooling power in the slow oscillation (0.5-1.5 Hz) and δ (1.5-4 Hz) frequency ranges. This makes a separate assessment of effect on slow oscillations and δ waves not possible. Electrodes used in this study were bundles implanted by hand, which is unlikely to lead to a reliable positioning in a defined cortical layer. In our work, all electrode positioning in the spindle-power rich deep cortical layers were verified post-hoc, and proper positioning was an important inclusion criterion. Finally, the effects of optogenetic stimulation of TRN cells reported by Lewis et al. are ambiguous: tonic TRN optogenetic stimulation actually decreased rather than increased firing rate in a majority of TRN cells (58.8%) in awake mice. Overall, the intended tonic excitation of TRN seems thus to have resulted in a substantial inhibition of these cells.

Together, we feel that it is premature to claim that the findings by Lewis et al. are inconsistent with ours. On the contrary, we argue that the predominant inhibition of TRN cells by optogenetic stimulation makes their observations possibly quite compatible with ours. In further support of this, it is mentioned in the text that spindle power actually decreases, an observation that we also make. However, likely differences in the precision of the recording sites, different analysis procedures and the use of different frequency bands makes final conclusions not possible at this point. Given these discrepancies, we prefer not to further discuss them in the main text.

7) Given the previous findings of TRN heterogeneity (Clemente, Halassa), the authors should close the loop, and determine whether the diversity in CaV 3.3 behavior they observe is actually related to the highly parallel diversity observed between PV and SOM cell types in the Clemente et al. work. There is a mixing of PV and other cell types in the recipient zones the authors identified: Slice recordings, for example, from a PV-tagged animal, and comparison of these cells in the SI pathway with all others, would likely be enough, though SOM tagged recordings would also be informative.

We fully agree with the reviewers that the parallels between the discharge patterns of PV- and SOM-expressing TRN cells and the ones of our sectorially identified cells are striking and should be clarified. This should ideally be done at all levels, from the in vitro analysis to sleep. Given that our paper studies sleep as the read-out for TRN heterogeneity, the principal and most forward moving question is to test whether the roles of PV- and SOM-expressing cells in natural non-REM sleep are indeed different. We thus carried out a series of sleep recordings in PV-Cre and SOM-Cre mouse lines (n = 8 each) that were chemogenetically manipulated in precisely the same way as the VGAT-Ires-Cre line in the original manuscript. The results obtained are presented in Author response image 1.

This first series of experiments indeed suggests that chemogenetic hyperpolarization of PV-, but not of SOM-expressing cells reproduces most observations found in the VGAT-Ires-Cre line, in which all TRN cells are hyperpolarized irrespective of their neurochemical nature. Thus, we find an increase in power in the δ (1.5-4 Hz) frequency band, while power in the slow oscillation (0.5-1.5 Hz) and the spindle bands (10-15 Hz) was reduced. Such changes could not be detected in the SOM-Cre expressing mouse line.

**Author response image 1. respfig1:** (**A**) Effects of chemogenetic hyperpolarization of PV-expressing TRN cells, experiment and analysis as described in Figure 4. (**B**) Effects of chemogenetic hyperpolarization of SOM-expressing TRN cells, experiment and analysis as described in Figure 4.

These data suggest that PV-expressing TRN cells are primarily implied in the control of local non-REM sleep. This result is consistent with the cellular and in vivo optogenetics findings described by Clemente-Perez et al. and with the important role of bursting cells for local non-REM sleep identified here. It is thus a starting point in bringing together neurochemical and electrophysiological heterogeneities.

Here are our arguments that make us think that these data are nevertheless preliminary and need important controls. First, the effects in the PV-Cre line are more moderate than in the VGAT-Ires-Cre line. This could mean that PV-expressing cells are only partially responsible for local sleep control, and that there other cellular subgroups involved. Alternatively, the chemogenetic manipulation in this mouse line could be weaker or less efficient. At the moment, we cannot distinguish between these possibilities. Moreover, the histological inspection of the brains used for these recordings make us think that the PV- and the SOM-expressing cell populations might not be completely separate and/or that there might be other cell groups. Additionally, we found virally driven mCherry expression in restricted cell groups of the dorsal thalamus in some mice. These were excluded from the data shown here but nevertheless made us doubtful about whether the findings can be unambiguously attributed to PV-expressing TRN cells. Immunohistochemistry in combination with viral tracing will be required to more definitely define the chemogenetically manipulated cell types and their roles in non-REM sleep.

We think this important and exciting next step requires a separate study combining optogenetically assisted circuit mapping and chemogenetics in in vitro and in vivo assessments. This goes beyond the scope of the current work. Therefore, in the revised manuscript, we define it as an important step to clarify in the second paragraph of the Discussion.

Minor point:The authors throw up their hands a little bit in the Discussion about how their algorithm might have impacted their findings. The authors should determine whether their algorithm impacts aspects of the behavior they observe. Specifically, they could sub-sample the spindles identified in the WT to be only from the same amplitude distribution as the Ca_V_3.3^-/-^ mice, and then see whether they still observe phase locking to the SO. This analysis should take just a few hours - they just need to limit their already conducted analyses to a smaller part of the data set - and would let them know much better if perhaps there is a fundamental mechanism at play.

We have carried out an analysis according to this excellent suggestion. Briefly, for every brain area we recorded, we selected spindle events in all wild-type animals that had an amplitude that fell within the mean ± 1SD of the amplitude of the spindle events in the Ca_V_3.3-KO animals in the corresponding brain area. The coupling of these remaining events to the slow oscillation was then again analyzed (See Author Response Image 2). The results show that the overall preferential occurrence of spindle events on the cortical upstate is preserved. This again suggests that the differential coupling between slow oscillations and spindles between WT and KO animals was not due to detection failures. To make this clear in the manuscript, we have added a sentence to the Results section “Ca_V_3.3 channels ensure the temporal coordination of sleep spindles with the active state of the SO”. This also has allowed us to shorten the Discussion, as requested in the point 3) of the essential revisions.

**Author response image 2. respfig2:** (**A**) Graph dots represent the amplitude of each spindle event of the WT detected during periods of slow oscillation SO (amplitude is extracted from the power of the FIR filtered signal in the 9-16Hz band) as a function of SO phase, for the 4 sites of recording, S1, S2, AC and PFC. Superimposed in grey is the normalized spindle occurrence (right y-axis) as a function of SO phase. Horizontal plain red line (and blue in the case of PFC) corresponds to mean amplitude of spindle events in the case of the Ca_V_3.3-KO, dashed line is the mean +/- 1 standard deviation. The dataset presented in this figure is the same as in Figure 6 of the main paper. (**B**) Sub-sampling of spindle events in the WT constrained by the amplitude of the Ca_V_3.3-KO for each area recorded: all spindle events from the WT outside the mean +/- 1SD from the Ca_V_3.3-KO are discarded. Superimposed in dark blue is the spindle occurrence (right y-axis) as a function of SO phase for the sub-sampled spindle events. (**C**) Mean occurrence of spindle onsets for the active (AS, -180°, 0°) and the silent (SS, 0°, 180°) state of the SO, in grey is the same WT data as in Figure 6, in dark blue is the sub-sampled WT data constrained to the Ca_V_3.3-KO mean +/- 1SD. D. Table presenting values of mean spindle amplitude and SD across Ca_V_3.3-KO mice and taken to apply the subsampling of WT spindle events as in **B** and **C**. Number of spindle events for each condition (across all WT mice) is indicated in the legend on the right.